# The Carbon Mapper emissions monitoring system

Riley Duren<sup>1</sup>, Daniel Cusworth<sup>1</sup>, Alana Ayasse<sup>1</sup>, Katherine Howell<sup>1</sup>, Alex Diamond<sup>1</sup>, Tia Scarpelli<sup>1</sup>, Jinsol Kim<sup>1</sup>, Kelly O'neill<sup>1</sup>, Judy Lai-Norling<sup>1</sup>, Andrew Thorpe<sup>2</sup>, Sander R. Zandbergen<sup>2</sup>, Lucas Shaw<sup>2</sup>, Mark Keremedjiev<sup>3</sup>, Jeff Guido<sup>3</sup>, Paul Giuliano<sup>3</sup>, Malkam Goldstein<sup>3</sup>, Ravi teja Nallapu<sup>3</sup>, Geert Barentsen<sup>3</sup>, David R.Thompson<sup>2</sup>, Keely Roth<sup>3</sup>, Daniel Jensen<sup>2</sup>, Michael Eastwood<sup>2</sup>, Frances Reuland<sup>4</sup>, Taylor Adams<sup>5</sup>, Adam Brandt<sup>4</sup>, Eric A. Kort<sup>5</sup>, James Mason<sup>3</sup>, Robert O. Green<sup>2</sup>

10


<sup>1</sup>Carbon Mapper, Pasadena, 91105, USA
<sup>2</sup>Jet Propulsion Laboratory, California Institute of Technology, Pasadena, 91109, USA
<sup>3</sup>Planet Labs PBC, San Francisco, 94107, USA
<sup>4</sup>Stanford University, Stanford, 94305, USA
<sup>5</sup>University of Michigan, Ann Arbor, 48109, USA






Correspondence to: Riley Duren (riley@carbonmapper.org)

## Abstract

The Carbon Mapper emissions monitoring system contributes to the broader ecosystem of greenhouse gas observations by locating and quantifying CH<sub>4</sub> and CO<sub>2</sub> super emitters at facility scale across priority regions globally and making the data accessible and actionable. The system includes observing platforms, an operational monitoring strategy optimized for mitigation impact, and a data platform that delivers CH<sub>4</sub> and CO<sub>2</sub> data products for diverse stakeholders. Operational scale-up of the system is centered around a new constellation of hyperspectral satellites. The Carbon Mapper Coalition (hereafter Tanager) satellites are each equipped with an imaging spectrometer instrument designed by NASA's Jet Propulsion Laboratory that are assembled, launched and operated by Planet Labs. The first Tanager satellite (Tanager-1) was launched 16 August 2024 completed commissioning in January 2025 and continued to improve observational efficiency through summer 2025. Planet is currently working to expand the constellation to four Tanagers. Each imaging spectrometer instrument has a spectral range of about 400-2500 nm, 5 nm spectral sampling, a nadir spatial resolution of 30 meters, and nadir swath width of about 19 km at the lowest orbital altitude. Each satellite is capable of imaging 250,000 km<sup>2</sup> per day on average. By combining the results of independent controlled release testing with empirical evaluation of the radiometric, spectral, spatial, and retrieval noise performance of the Tanager-1 spectrometer, we predict minimum detection limits of about 64 – 126 kgCH<sub>4</sub>/h for CH<sub>4</sub> point sources and about 10,078 – 18,994 kgCO<sub>2</sub>/h for CO<sub>2</sub> point sources for images with 25% albedo, 45 degree solar zenith angle, and 3 m/s wind speed. A review of the first 11 months of Tanager-1 CH<sub>4</sub> and CO<sub>2</sub> observations including initial validation with coordinated aircraft under-flights and non-blind controlled release testing indicates that the system is meeting performance requirements and, in many cases, surpassing expectations. We also present early evaluations in challenging onshore and offshore observational conditions and summarize the first use of Tanager data to guide the timely mitigation of a CH<sub>4</sub> super emitter.




#### 1 Introduction and objectives

As governments, companies and actors across civil society pursue a broad range of efforts to stabilize and reduce greenhouse gas (GHG) emissions there is an increasing need for actionable emissions data that is accurate, timely, and trusted. The expanding portfolio of use-cases includes diverse governmental regulations, private sector market-based initiatives, leak detection and repair programs, and an increasing demand for measurement-based emission inventories and enhanced transparency. Example includes ambitious emission mitigation targets by governments under the Paris Agreement (UN 2015), Kigali Amendment (UN 2016), and Global Methane Pledge (UN 2023) as well as major private sector initiatives such as the Oil and Gas Methane Partnership (UNEP 2020). In parallel with these policy developments, dramatic advances in GHG measurement technology have occurred over the last several years – particularly for methane (CH<sub>4</sub>) and carbon dioxide (CO<sub>2</sub>), the top two climate pollutants.

Scaling up a global emissions monitoring system that can provide global operational tracking of millions of emission sources at facility-scale along with rapid, transparent data publication requires addressing technical as well

as institutional barriers. To confront these challenges, the Carbon Mapper non-profit organization (https://carbonmapper.org/) was established with the support of philanthropists to provide leadership in observational system development, CH<sub>4</sub> and CO<sub>2</sub> science data analysis, and stakeholder engagement. We also assembled a public-private partnership called the Carbon Mapper Coalition to design and launch the first in a series of next generation satellites. The Carbon Mapper Coalition (hereafter Tanager-series) satellites are each equipped 60 with an imaging spectrometer instrument designed by NASA's Jet Propulsion Laboratory (JPL) that are assembled, launched and operated by Planet Labs. Like existing hyperspectral research satellites (e.g., PRISMA, EMIT, and EnMAP), Tanager is multi-application ready. However, unlike those missions, for the sake of operational GHG monitoring globally, a large portion of Tanager's tasking capacity is allocated to mapping known CH4 and CO2 emitting regions or infrastructure. The first Tanager satellite (Tanager-1) was launched 16 August 2024, completed 65 commissioning in January 2025, and is on track to begin full operational monitoring by summer 2025. In this paper we describe the design, observational strategy, and performance of the Carbon Mapper emissions monitoring system as enabled by a constellation of Tanager satellites.

#### 1.1 Motivation and challenges for measuring and mitigating CH<sub>4</sub> and CO<sub>2</sub> point sources










Since 2005, the ability to quantify global GHG concentrations using backscattered solar radiance at various spatiotemporal scales has existed from space with atmospheric sounding satellites, which for CH4 began with SCIAMACHY (Frankenberg et al., 2011) and more recently GOSAT (Turner et al., 2015) and the TROPOMI instrument onboard the Sentinel-5p satellite (Hu et al., 2018). CO2 measuring satellites include NASA's OCO-2 and OCO-3 missions (Crisp et al., 2004; Eldering et al., 2019) as well as GOSAT. These early generations of atmospheric sounding satellites - measure gas absorption features at sub-nanometer spectral resolution with high precision, but doing so requires coarse, multi kilometer-scale spatial resolution (Jacob et al., 2022). These systems are optimized for quantifying the total CH<sub>4</sub> and CO<sub>2</sub> fluxes for large regions, including the net contributions of diffuse area sources (typically distributed over several kilometers) and condensed point sources (typically originating from surface features < 10 meters across). Therefore, detection and quantification for individual point sources from these instruments is only sometimes possible for very large sources whose emission are generally more than 8t CH<sub>4</sub>/h (Schuit et al., 2023) to 50 t CH<sub>4</sub>/h (Lauvaux et al., 2022) or 1600 t CO<sub>2</sub>/h (Nassar et al., 2021). The regional flux mapping satellites have provided important constraints for atmospheric GHG budgets when their observations are assimilated with atmospheric chemistry and transport inverse models (e.g., Worden et al., 2022, Byrne et al., 2022). However, the spatial resolution of global inverse models is typically on the order of 25 - 400 km, which may be sufficient for regional flux quantification, but not identifying and quantifying individual point sources at the scale of meters or capturing the bulk of point source distributions. Past inverse modeling work has attempted to estimate facility-scale emissions for spatially isolated sources like individual landfills (e.g., Nesser et al., 2024), but this requires temporal averaging, and even average emissions cannot be attributed to facility-scale for dense infrastructure regions like oil/gas fields or urban areas. MethaneSAT was launched in March 2024 and until June 2025 quantified both the total CH<sub>4</sub> emissions and larger point sources for global oil and gas production basins with significantly higher spatial resolution than other regional flux mappers (Miller et al., 2024). Most recently, the Japanese space agency launched the GOSAT-GW satellite in June 2025 which is currently undergoing commissioning and is designed to provide global CH4 flux mapping with a 3 day revisit at spatial resolutions ranging from 1 to 10 km (Tanimoto et al., 2025).

Meanwhile, multiple studies have conclusively identified the existence of methane "super emitters" where a relatively small fraction of infrastructure is often responsible for a disproportionate fraction of total emissions from key regions and economic sectors. CH<sub>4</sub> super emitters have the potential for emission rates exceeding 100 kg/h and are often associated with point sources. Super emitters can be the result of leaks, malfunctioning equipment, or process venting – many of which can be temporary but, in some cases, may persist for months to years (Cusworth *et al.*, 2024). Beginning in 2016 and 2017, advanced remote-sensing aircraft were used to conduct the first comprehensive, economy wide survey of methane emitters in California and found that less than 0.2% of the infrastructure is responsible for over a third of the state's entire methane inventory (Duren *et al.*, 2019). Since then, these intensive field campaigns have expanded to other key regions across the US and other jurisdictions and show that a relatively small fraction of facilities are responsible for 20-60% of total emissions spanning multiple economic sectors (Cusworth *et al.*, 2022; Sherwin *et al.*, 2024). Additionally, since 2020, Carbon Mapper aircraft surveys of California, Colorado, Pennsylvania, New Mexico and Texas have demonstrated that delivering actionable emissions data to facility operators and agencies can lead to expedited mitigation action (CARB 2022; CDPHE 2024). Participating operators have reported that roughly half of the methane emissions we identified at their facilities

(primarily from oil and gas production, downstream natural gas and solid waste management sectors) are "fixable" and in many cases we have verified emission reductions with follow-up overflights. Another key finding is that many of these super emitters are highly intermittent, widely dispersed and difficult to find with conventional surface measurements that have limited coverage due to cost and logistical constraints [Cusworth *et al.*, 2021a].





Given the major contribution of super emitters to regional methane budgets and the opportunity they present for mitigation, tools are needed that can provide global monitoring of these large, dispersed and often transient sources. Aircraft and satellite instruments optimized for point source imaging can complement area flux mapping satellites by quantifying emissions from point sources at very high (1-30 meter) spatial resolution. Unlike area flux mapping satellites optimized for high spectral sampling over a narrow spectral range, many point source imagers tend to have coarser spectral sampling (5-10 nm) while being sensitive to the full Visible to Shortwave Infrared (VSWIR) spectral range of solar backscatter (e.g., 400-2500 nm) - which allows for a broader range of applications beyond trace gas sensing. Field campaigns with a class of VSWIR imaging spectrometers such as the next generation Airborne Visible/Infrared Imaging Spectrometer (AVIRIS-NG) and Global Airborne Observatory (GAO) have routinely shown that when flown at altitudes of 3 - 8km, single point source CH<sub>4</sub> emissions above 5-10 kgCH<sub>4</sub>/h for 3 m/s wind can be detected and quantified, and their plumes can be mapped with 3-8 m spatial resolution (Frankenberg et al., 2016; Duren et al., 2019; Cusworth et al., 2021a). Those airborne instruments have also demonstrated the ability to quantify CO<sub>2</sub> point source emissions as low as about 8,000 kgCO<sub>2</sub>/h (Kim et al., 2025) for a 3 m/s wind speed. The major benefit of such instruments is that their high spatial resolution can enable accurate attribution of observed CH<sub>4</sub> and CO<sub>2</sub> to plumes to specific emission sources at (and sometimes within) individual facilities. A key limitation of point source imagers is that they are primarily sensitive to discrete point sources rather than diffuse area sources and net emissions from regions. Another limitation of point source imagers is that singular instruments have limited spatio-temporal coverage, particularly aircraft surveys that are limited by logistics, cost and airspace restrictions. Individual point source imaging satellites can provide greater spatiotemporal coverage than aircraft but are still limited compared to regional flux mapping satellites. It is through the coordinated observations and analysis of data from point source imagers and regional flux mappers that truly complete, multi-scale understanding of CH<sub>4</sub> and CO<sub>2</sub> emissions can be obtained, have been demonstrated in previous studies (Cusworth et al., 2022; Naus et al., 2023).




135

## 1.2 Other point source imaging satellites

Building on the success of airborne campaigns, there have been considerable advances with point source imaging satellites over the past several years. GHGSat is a company that currently operates a constellation of 12 cubesats that were the first satellites to offer operational monitoring of CH4 point sources as a commercial service. Each GHGSat uses a novel Fabry-Perot spectrometer that is optimized for imaging facilities and small areas (~ 12 km x 12 km) at 25 m resolution (Jervis et al 2021). GHGSat reports a 50% probability of detection of 120 kgCH4/h (Jervis et al., 2022) for a 3 m/s wind speed although independent studies suggest more typical detection limits of about 180 kgCH4/h and 2.1% CH4 single measurement precision for average brightness scenes (McLinden et al., 2024). Although it is designed to map emissions at known facilities with rapid revisit, GHGSat is not optimized for mapping large regions because the retrieval method requires many samples over a given target to derive a single methane image.

Multi-spectral land imagers such as Sentinel-2 and Landsat-8 have also demonstrated some CH<sub>4</sub> point source detection capability however their high spatial resolution (20 and 30 m, respectively) is offset by much coarser spectral resolution (200 nm FWHM) which translates to single measurement precisions >30% (Jacob *et al* 2022).

A broader general class of point source imagers are VSWIR imaging spectrometers. As with the AVIRIS series of airborne sensors, most current satellite VSWIR spectrometers are hyperspectral instruments designed primarily to address a large range of earth science research topics spanning terrestrial ecology and geology. These systems typically involve relatively high spatial resolution (30 - 60 meters), moderate signal to noise ratio, and moderate spectral resolution. While swath widths of this class of instruments are typically narrower than regional flux mappers, they tend to image relatively long strips due to a pushbroom mode of operation and hence can be efficient at mapping larger regions at high spatial resolution. Most of these missions were not designed specifically to detect CH<sub>4</sub> but many have demonstrated varying degrees of capability. For example, PRISMA, EnMAP and Gaofen-5 all report 30 meter spatial resolution and spectral resolution of 10 nm full width half maximum (FWHM)

(Guanter et al 2023, Roger et al 2024, Irakulis-Loitxate *et al.*, 2021). The single measurement CH<sub>4</sub> precisions of these instruments has been estimated to be 3-9% (Jacob et al., 2022). NASA's EMIT instrument has 60 meter spatial resolution, spectral resolution of about 9 nm FWHM, and estimated single measurement precision of 2-6% (Jacob *et al.*, 2022). Given their specifications, PRISMA, EnMAP Gaofen-5, and EMIT are anticipated to have 90% probabilities of detection for CH4 point sources of 1000 kgCH<sub>4</sub>/h or greater, however independent evaluations are still underway.

As described below, the Carbon Mapper monitoring system is designed to contribute to the growing ecosystem of methane observations by combining regional coverage, high precision, and ultimately high frequency sampling. This is enabled by the Tanager satellite constellation with higher spectral resolution and signal to noise ratio than most existing VSWIR imaging spectrometers.

#### 1.3 Carbon Mapper monitoring system overview and objectives

Carbon Mapper has a public good mission to enable GHG emission reductions by making methane (CH<sub>4</sub>) and carbon dioxide (CO<sub>2</sub>) data more complete, actionable and accessible. We contribute to the emerging multi-scale ecosystem of global GHG observations by locating and quantifying CH<sub>4</sub> and CO<sub>2</sub> point sources at facility scale across key regions. While other programs are optimized for quantifying wide area methane emissions at the scale of regions and major oil and gas production basins, Carbon Mapper provides high resolution and high frequency tracking of methane and CO<sub>2</sub> emissions of individual facilities and pieces of equipment globally. Our primary objectives are to provide actionable mitigation guidance to facility owner/operators and regulators and improve awareness of emissions across civil society. Carbon Mapper data also can be integrated with other data sets to help evaluate and improve greenhouse gas inventories and accounting frameworks in support of the Global Methane Pledge, Paris Climate Agreement, and mitigation targets of key sub-national jurisdictions. The Carbon Mapper emissions monitoring system is designed to fill critical gaps in the emerging global framework of greenhouse gas observing systems, providing actionable data at facility scale to drive leak repair efforts and hold emitters accountable. Stakeholders include US federal and state agencies and their counterparts in other countries, international data programs and registries, facility operators, development banks, non-governmental organizations, and civil society.

The Carbon Mapper emissions monitoring system includes observing platforms (Tanager satellites, aircraft, and EMIT), an operational monitoring strategy optimized for mitigation impact, and a data platform that delivers actionable, accessible, and transparent CH<sub>4</sub> and CO<sub>2</sub> data products for diverse stakeholders. Global scaling of this system is centered around the new constellation of Tanager satellites. The design of those satellites, along with Carbon Mapper's strategy for emissions monitoring and data platform were informed by campaigns and mitigation pilot projects using prototype aircraft sensors.

#### 2 System Design











Carbon Mapper's emissions monitoring system is motivated by an overarching goal to make CH<sub>4</sub> and CO<sub>2</sub> super emitters visible and to deliver data to guide efforts to mitigate (eliminate or reduce) them. The system design and operations are in turn driven by our priorities of maximizing completeness, actionability, and accessibility. Completeness (also called Observing System Completeness) to be the percentage of a given population of emitters that can be detected based on an optimal balance of detection limits, spatial coverage and sample frequency (Jacob *et al.*, 2022). Actionable means timely data delivery and notification (e.g., latencies measured in hours and days rather than months) with precise and reliable geolocation and attribution of observed CH<sub>4</sub> and CO<sub>2</sub> plumes to specific emissions sources. Accessible means that data is available to the largest possible set of stakeholders, is transparent and in formats that are readily understandable by a wide audience.

Carbon Mapper's monitoring strategy and the Tanager satellites are designed to optimize completeness. This metric constrains ultimate mitigation potential because characterizing a critical set of emitting infrastructure requires routine observation across large areas to identify specific leaks, equipment malfunctions and inefficient process venting. Actionability is the second major design driver. Using remote sensing to guide CH<sub>4</sub> leak detection and repair (LDAR) action requires that high emission events can be detected and reported quickly enough so that facility

operators can verify and diagnose the root cause with follow-up site visits. Some super emitter events have the potential to eclipse the normal net annual emissions of an entire facility within a few days or weeks if not detected and repaired in a timely fashion (Pandey et al., 2019). The Carbon Mapper emissions monitoring system is designed to deliver actionable information - images of emission plumes, estimated source coordinates, emission rate estimates, sectoral attribution - within 72 hours of each observation. The low latency capability is enabled by the Planet's small satellite platform that includes a high-speed downlink and backhaul capability combined with low latency data processing by the Planet and Carbon Mapper data platforms. The ability to precisely and reliably geolocate and attribute observed plumes to a physical emission source is critical both for effective support of LDAR programs and quantification of high emission activity at facility-scale. Another major design driver is data accessibility, where transparent availability of quality-controlled CH4 and CO2 emissions data is intended to provide maximum support for monitoring, reporting and verification programs, measurement informed inventories, and improved situational awareness for a diverse audience of stakeholders.

## 2.1 Emissions monitoring strategy









Mitigation potential is ultimately constrained by the fraction of emissions from a given population that can be observed. Our system is designed to provide sustained operational monitoring of the world's CH4 and CO2 super emitters at facility scale. In doing so, our program complements and leverages other key CH4 and CO2 observing systems, such as S5/TROPOMI, GOSAT-GW, OCO-2, OCO-3, etc., that provide critical insights into net regional emission fluxes including the contributions of diffuse area sources. At the same time, a key element of our strategy is scalability and continuity of the Tanager constellation which is strongly dependent on available financing for assembly, launch and sustained operation of the satellites. This is enabled by Planet's use of Tanager to support a broad portfolio of environmental indicators beyond CH4 and CO2 (e.g., various land surface variables) that are derived from hyperspectral imaging across the full VSWIR spectral range. As a result, a large fraction of Tanager tasking capacity is allocated to observing known CH4 and CO2 emitting regions identified by Carbon Mapper and the remainder is allocated to either commercial CH4 services, other hyperspectral applications, and/or maximizing coverage of land areas. A general operating rule for Tanager is to "always be imaging" – meaning the satellite will image targets of opportunity over land when not in conflict with other tasking priorities or completely overcast conditions. Regardless of which application motivates tasking of a given area, Carbon Mapper processes every Tanager image to quantify and publish any CH4 and CO2 emissions that are detected.

There are no formal specifications of super emitter populations that span all emission sectors and processes, but to guide our strategy we have established a reference distribution of emitters informed by empirical field campaigns, emission inventories, and analysis of Geographical Information System (GIS) datasets. For CH4, we use findings from aircraft remote sensing surveys covering over half of US oil and gas production and over 250 landfills across multiple regions, jurisdictions, and time scales (Duren *et al.*, 2019, Cusworth *et al.*, 2024). The aircraft measurement methods used in those studies typically offer a 90% probability of detection limit of 10-45 kg/h for 3 m/s wind speeds based on single-blind controlled release testing (El Abbadi *et al.*, 2024; Ayasse *et al.*, 2023). Additionally, a synthesis analysis combined nearly 1 million empirical measurements from those and other aircraft surveys with similar detection limits to construct a statistical model of all emissions > 0.1 kg/h for major US oil and gas basins and found that 20-80% of total emissions come from a relatively small population of sources emitting > 100 kgCH4/h (Sherwin *et al.*, 2024). Hence, we set 100 kgCH4/h as a reference definition for a CH4 super emitter. The same threshold was adopted by the US Environmental Protection Agency's Super Emitter Program and Greenhouse Gas Reporting Rule for the oil and gas sector (US EPA 2024a; US EPA 2024b). Similarly, for CO<sub>2</sub>, previous empirical studies and emission inventories indicate that 90% of fossil fuel power plant emissions come from plants emitting > 100,000 kgCO<sub>2</sub>/h (Cusworth *et al.*, 2021b). Hence, we adopt this as our definition of a CO<sub>2</sub> super emitter.

Effective emissions monitoring strategies rely on multiple variables and so to provide quantitative guidance for optimization, we apply the Observing System Completeness (C) metric, defined as the fraction of a reference population of emitters that can be detected by a constellation of satellites as a function of detection limit (C<sub>D</sub>), spatial coverage (C<sub>S</sub>), and temporal sampling (C<sub>T</sub>) (Jacob *et al.*, 2022).

 $C = C_D \times C_S \times C_T \tag{1}$ 

Providing a detailed treatment of Completeness, particularly a robust treatment of the analysis that addresses satellite constellation design to optimize spatial coverage and temporal sampling in the presence of considerable spatio-temporal variability in emission source populations, is beyond the scope of this paper and will be covered in a separate manuscript. However, because C<sub>D</sub> is driven by single satellite performance we elaborate on it here.

C<sub>D</sub> describes the fraction of a given population that exceeds the detection limit of the instrument and retrieval algorithms. CD is constrained by spatial resolution and single measurement precision for a specified emissions distribution and set of environmental conditions - primarily albedo, solar zenith angle, and wind speed. To achieve an optimal CD we use the findings from the field studies described above to set a goal of 90% empirical probability of detection (POD) for point sources ≥100 kg/h for CH<sub>4</sub> and ≥100,000 kg/h for CO<sub>2</sub>. A related approach for specifying detection limits is to define a Minimum Detection Limit (MDL) - a flux above which a detection could be considered confident. Historically MDL has been calculated analytically from the measurement noise based on key instrument parameters such as spectral, radiometric, and spatial performance. The simplicity of this calculation made it helpful in defining system requirements. Here, in contrast, our empirical POD standard is calculated from controlled release experiments. This makes it a more robust metric as it accounts for not just the instrument design on a per-pixel basis, but also the real-world imaging conditions with complications such as turbulence, surface clutter, and other effects that are difficult to model, along with the complete observational and analysis workflow. The empirical POD is more representative of the range of actual detection rate under real world conditions. However, it requires empirical evaluation of an as-built instrument and data analysis system spanning a range of emission rates using controlled release experiments and/or coordinated observations of a population of sources with independent measurement systems. We are currently working to establish an empirical 90% POD for Tanager methane data using a variety of methods which is anticipated to require a full year to collect enough coordinated observations. Meanwhile, to guide Tanager requirements during design and development we used airborne data collected from past field experiments to establish a linear relationship between the empirical 90% POD and analytical MDL following methods described by Ayasse et al., 2023. For a 90% empirical POD of 90 kgCH<sub>4</sub>/h we estimated an equivalent MDL of 63 kgCH<sub>4</sub>/h for Tanager's highest sensitivity imaging mode. That MDL target was used to set single measurement precision requirements and relate them to key instrument parameters such as spectral sampling, SNR, and spatial resolution. In addition to instrument design, SNR also varies with environmental factors such as surface albedo and solar zenith angle, both of which in turn can vary with latitude and season. Our analysis and on-orbit validation of single measurement precision and MDL is described in section 4.

310

330







#### 2.2 Instrument

Since 2016 studies with aircraft imaging spectrometers such as the Airborne Visible/Infrared Imaging Spectrometer 315 (AVIRIS) series and Global Airborne Observatory (GAO) as well as research satellites have led to the publication of over 15,000 CH<sub>4</sub> plumes to date (e.g., Thompson et al. 2015, Frankenberg et al. 2016, Duren et al. 2019, Cusworth et al. 2022, Thorpe et al. 2023, Cusworth et al. 2024) as well as initial CO2 point source studies (Thorpe et al. 2017, Cusworth et al. 2023). Pilot projects have used AVIRIS-NG and GAO and, more recently, analysis of data from NASA's EMIT (Earth Surface Mineral Dust Source Investigation) instrument on the International Space Station (ISS) 320 - to demonstrate the utility of this technique for actionable geolocation and quantification of methane point source emissions. However, none of those instruments were designed for greenhouse gas sensing. Deploying a truly operational point source imager in space with improved area coverage while minimizing CH<sub>4</sub> and CO<sub>2</sub> detection limits introduced some design challenges, e.g. the instrument must provide increased light gathering power without sacrificing spectral sampling or spectral range. Preserving the full VSWIR spectral range rather than developing a 325 spectrometer narrowly focused on the SWIR bands is necessary given the need to underpin the expansion and continuity of the Tanager constellation with commercial revenue for environmental data products that extend beyond CH<sub>4</sub> and CO<sub>2</sub> to serve a wide range of land and ocean hyperspectral applications.

Practically, this means that each Tanager imaging spectrometer instrument must have a detector sensitive to a wide spectral range, with large pixels which can efficiently collect many photons. It must also have a small focal ratio, or f-number, for maximum light-gathering power and high signal to noise ratio (SNR). It must have a fine spatial resolution, so that individual CH<sub>4</sub> or CO<sub>2</sub> emission plumes can be geolocated precisely enough for facility operators to quickly find and verify with follow-up site visits (e.g., within about 30 meters). The instrument must also have a

sufficiently wide swath and along-track imaging capability to efficiently map large oil and gas production fields, major urban areas, and other priority regions for  $CH_4$  and  $CO_2$  point source emissions that can occur in a stochastic fashion. The key instrument parameters that resulted from these design trade-offs are summarized in Table 1. The swath width values shown here are for the final operational orbit altitude of 406 km; these values are  $\sim 30\%$  higher during commissioning operations for a typical initial injection orbit of 510 km. Currently, the average orbit altitude is 430km and the swath width is approximately 20 km.

**Table 1.** Imaging spectrometer instrument specifications for a 406 km orbit altitude.

| Parameter                           | Value                                        |
|-------------------------------------|----------------------------------------------|
| F-number                            | F/1.8                                        |
| Focal Length                        | 400 mm                                       |
| Entrance Pupil Diameter             | 224.5 mm                                     |
| Pixel Size                          | 30 μm                                        |
| Detector Array Size                 | 640 x 480 pixels                             |
| iFOV                                | 75.3 urad                                    |
| Cross-track FOV                     | 45.9 mrad                                    |
| Swath Width                         | 18.6 km (nadir) – 26.4 km (30 deg off-nadir) |
| Spectral Range                      | 380 - 2500  nm                               |
| Spatial Resolution                  | 30 meters (nadir)                            |
| Spectral Sampling                   | 5 nm                                         |
| Spectral Response (FWHM)            | ≤6 nm                                        |
| Signal to Noise Ratio (SNR)         | 400-800 @ 2300 nm                            |
| Spectrometer Temperature            | 250 K                                        |
| Focal Plane Array Temperature       | 160 K                                        |
| Radiometric Calibration Uncertainty | ≤10%                                         |
| Spectral Calibration Knowledge      | 5%                                           |
| Spectral Cross-Track Uniformity     | 2%                                           |
| Instrument mass                     | 78 kg                                        |

The resulting Tanager instrument design leverages four iterations of previous imaging spectrometer development and refinement by the Jet Propulsion Laboratory, including instruments such as the Airborne Visible/Infrared Imaging Spectrometer (AVIRIS) series of airborne instruments, the Moon Minerology Mapper, and most recently, the EMIT instrument on the International Space Station. EMIT achieved first-light on 28 July 2022, with excellent uniformity and calibration, meeting all performance requirements (Thompson *et al.*, 2024). With a 60 meter spatial resolution, 7 nm spectral sampling, low inclination orbit and no ground motion compensation, EMIT is not optimized for greenhouse gas monitoring. However, EMIT has already provided a demonstration of CH<sub>4</sub> and CO<sub>2</sub> plume detection and quantification (Thorpe *et al.*, 2023) and Carbon Mapper's data platform has been routinely publishing those products to exercise workflow and quality control procedures in preparation for the Tanager satellites.

The instrument design is summarized in Supplemental Information section SI-1. Instrument specifications are summarized in Table 1 and have been validated by a combination of lab tests and on-orbit measurements. The instrument design achieves excellent spectral uniformity, with keystone and smile each 

The instrument is hosted by Planet's Tanager smallsat platform which provides power, precision pointing and high speed data storage and downlink, further described in Supplemental Information section SI-2.

#### 2.3 Orbit and Imaging Modes









Each Tanager satellite is launched into a near polar Sun Synchronous Orbit (SSO) with a goal of achieving an optimal local time of the descending node (LTDN) or crossing time at the equator between 1100 and 1300 hours. This ensures consistent and maximum daylight illumination for spectroscopic observations. The initial injection altitude varies by individual launch but is typically around 510 km. Each Tanager satellite uses electric propulsion to maneuver to the final operational altitude of 406 km and then maintains operational altitude over the 5-year design life. For example, Tanager-1 was launched into an initial 510 km altitude orbit with an initial LTDN of approximately 1040 hours. Tanager-1 was subsequently maneuvered into an interim orbit with a 430 km average altitude by January 2025 with a goal to complete maneuvering to the target 406 km and 1200 LTDN later. The final orbit parameters are selected to provide an optimal balance between SNR, spatial resolution, FOV and global access. By deploying multiple satellites with differing orbital planes and crossing times we can better constrain the variability and intermittency of emission sources.

The observing geometry of each Tanager satellite is described in Fig. 1. The imaging spectrometer is a pushbroom sensor where the image swath width is set by the instrument iFOV, 640 cross-track (spatial) elements of the focal plane, and the degree of off-nadir pointing. Our primary observing strategy leverages the agility of the Tanager satellites to provide cross-track imaging of strips, nominally up to  $\pm$  30 degrees left or right of the nadir ground track. This is done by rolling the satellite to a fixed off-nadir angle prior to imaging. The length of each image is set by the imaging mode.

As summarized in Fig. 2 and table 3, Tanager offers four standard imaging modes ranging from 1 to 4 exposures per surface footprint. Tanager satellites offer ground motion compensation (GMC) by back-nodding in the along-track direction to offset the orbital rate during an image exposure. This prevents along-track pixel stretch while enabling multiple exposures of a surface footprint. Since the instrument is shot noise limited, the effective SNR scales with  $\sqrt{N}$  where N is the number of exposures. For example, using the instrument standard 8 millisecond integration time per exposure, GMC allows up to 4 exposures to be acquired of the same surface footprint, where image stacking results in an effective integration time of about 32 ms and SNR about twice that of a single 8 ms exposure. Increasing the number of exposures increases SNR at the expense of decreasing along-track strip length and hence image area. Single image areas range from 346 to 8,240 km². This provides flexibility for trading off detection limit versus area coverage when designing an observing strategy for a given region, emission sector, or stakeholder usecase. Similar to EMIT and other airborne and satellite imaging spectrometers (but not shown in Fig. 2), the Tanager satellites are also capable of operating in pure nadir-viewing push-broom mode that allows image captures as long as 1080 km but doing so results in pixels that are 60 meters along track, with lower effective SNR, and less efficient sampling.

Figure 1. Observing geometry. (Left) Conceptual illustration of the instrument Field of View (FOV). The swath width is set by the focal plane array's 640 cross track spatial pixels projected onto the earth below the satellite. The spectrometer disperses light across the 480 spectral pixels in each line as the satellite forward motion images the earth. (Right) Top-down view illustrating Tanager's ability to roll in the cross-track direction (nominally up to  $\pm$  30 degrees) to image strips to the left or right of the nadir ground path.

**Figure 2.** Tanager standard imaging modes for 406 km altitude orbit. For each mode, the Tanager satellite backnods with a given angular rate over some distance to provide ground motion compensation. The top part of the figure indicates satellite orientation at the start, middle and end of an image acquisition. The bottom part of the figure indicates the footprint of a single pixel for the specified number of 8 millisecond exposures. The range of imaging modes provides flexibility to trade between along-track spatial coverage and effective integration time for each image by selecting varying degrees of ground motion compensation. From left to right: standard sensitivity (single exposure) with 443 km maximum image strip length, medium sensitivity (two stacked exposures) with 146 km maximum strip length, high sensitivity (three stacked exposures) with 87 km maximum strip length, and maximum sensitivity (4 stacked exposures) with 62 km maximum strip length. Single image areas range from 346 to 8,240 km² and the along-track spatial resolution is preserved in each of these modes. The maximum strip lengths and areas shown here increase by about 15-20% for a 430 km altitude orbit.

**Table 3.** Summary of integration time, spatial coverage, and SNR in the 2300 nm methane band (assuming a 25% albedo and 35 degree solar zenith angle) for the primary imaging modes at an altitude of 406 km. For each mode, area coverage is adjustable by selecting the along-track image length. Standard (1x8) imaging mode is planned for most ocean glint observations however the maximum image area will typically be constrained by the size of the sun glint spot.

| Imaging<br>Mode<br>(Sensitivity) | N samples x Integration time (ms) | Minimum Area per<br>Image (km²) | Maximum Area per<br>Image (km²) | SNR at 2300<br>nm |
|----------------------------------|-----------------------------------|---------------------------------|---------------------------------|-------------------|
| Maximum                          | 4 x 8                             | 346                             | 1,153                           | 615               |
| High                             | 3 x 8                             | 346                             | 1,618                           | 532               |
| Medium                           | 2 x 8                             | 346                             | 2,716                           | 435               |
| Standard                         | 1 x 8                             | 346                             | 8,240                           | 307               |

Given that 30% of global oil and gas production occurs in offshore environments any complete monitoring system must be capable of assessing emissions from production platforms, drilling ships and related infrastructure.

Previous aircraft studies indicate that the magnitude, intensity and persistence of CH<sub>4</sub> emissions in some offshore environments exceed what is routinely seen in onshore production (e.g., Biener *et al.*, 2024, Gorchov Negron *et al.*, 2023, Ayasse *et al.*, 2022). Tanager satellites are designed to conduct sun-glint observations over the ocean. This is critical for detecting methane emissions from offshore oil and gas platforms and potentially tanker vessels given that the ocean is dark at SWIR wavelengths. Aircraft prototyping indicates that sufficiently agile platform pointing and advance planning can align the instrument line of site with the bright specular sun-glint spot and infrastructure of interest (Ayasse *et al.*, 2022). Tanager sun-glint observations typically use Standard imaging mode. In addition to the normal constraints for task planning for onshore targets, specular conditions for glint additionally require that the following two geometrical conditions are satisfied. 1) The elevations subtended by the spacecraft and the Sun are equal. 2) The Sun and Satellite are on the opposite sides of the target, i.e., the relative azimuth between the satellite and sun is 180 degrees. The number of imaging opportunities for ocean glint spots is more restricted than for land observations given the need to align the sun-glint spot over areas of interest. For example, a maximum duration land observation in Standard imaging mode is 126 seconds or 443 km along track but most sun-glint observations will be limited to about 53 km along track (Nallapu *et al.*, 2022). While Tanager-1 is demonstrating the successful

Following launch, each Tanager satellite undergoes a commissioning phase that lasts several months and includes activation and checkout of spacecraft and instrument subsystems, a first light campaign to evaluate initial spectrometer performance at the injection altitude, followed by several months of propulsive maneuvers into the final lower orbit where calibration and validation efforts are completed prior to transition into steady state operations.

application of sun-glint observations, we have not yet transitioned to operational offshore mapping.

#### 2.4 Data platforms and products

- The focus here is on data processing applied to Tanager observations however the same basic procedures for generating calibrated radiance (Level 1) products are applied to the aircraft and EMIT observations used by the Carbon Mapper emissions monitoring system. Carbon Mapper retrievals, plume detection, emission estimation and quality control procedures are also uniformly applied to Tanager, EMIT and aircraft data to generate Level 2 4 CH<sub>4</sub> and CO<sub>2</sub> data products. Figure 3 summarizes the operational data analysis workflow for CH<sub>4</sub> and CO<sub>2</sub> emissions.
- For Tanager observations, Planet's data platform generates the Level 1 data product that consists of georectified radiance files that are delivered to Carbon Mapper's data platform for retrieving CH<sub>4</sub> and CO<sub>2</sub>. This process begins with Planet's data platform receiving raw data from the Tanager satellites via the Smallsat Platform high speed downlink and backhaul. The raw images are first orthorectified using a combination of digital elevation data and a full state model that is iteratively refined using a set of globally distributed ground-controlled points derived from reference imagery (e.g., Landsat 8). The combination of this approach with pointing telemetry from the spacecraft enables rectification accuracy well below the 30 meter (CE90) requirement.
- Planet conducts pre- and post-launch calibration of each payload to quantify spectral, radiometric, spatial and uniformity characteristics. The calibration and characterization of the Tanager instruments is based on procedures developed by JPL that have been refined and proven over more than 30 years including those successfully demonstrated with the AVIRIS series of airborne instruments and the EMIT mission (Thompson *et al.* 2024). These procedures allow raw sensor digital numbers (DN) to be converted to physical sensor units. The result is a calibrated
   Top Of Atmosphere (TOA) radiance hypercube for each image. Additionally, the Planet data platform generates a cloud mask to support Level 2 processing. Level 1 processing is further described in Supplemental Information section SI-3.

Figure 3. Carbon Mapper data analysis workflow applied to Tanager, EMIT and aircraft observations.

For Tanager observations, the Carbon Mapper data platform uses calibrated radiance files delivered by Planet to derive several Level 2 data products. The L2 image outline products are the geographic boundaries, or "strips" of areas imaged by the Carbon Mapper Coalition satellites. Strip image outlines are helpful for determining where data is collected, the quality of that data, and verifying when methane or carbon dioxide sources are imaged, but no emissions above our detection limit were observed. In optimal observing conditions, such as an unobstructed view of the emission source and a high likelihood of detection, the absence of detection is termed a "null detect." The null detects imply that the source is not emitting methane above the sensor's minimum detection limit. We consider an image to be a good candidate for a null detect status for an emission source if the image contains less than 25% cloud cover and intersects any of the plume origin points estimated for the source. All Tanager L2 products are resampled to 30 meter resolution. A summary of key Level 2-4 CH<sub>4</sub> and CO<sub>2</sub> data analysis steps is provided here with more detailed information in Supplemental Information section SI-4.

## L2A Reference basemap images

L2A products are three-band (red-green-blue), natural color images of the Earth's surface generated from Tanager radiance files. This process involves correcting for atmospheric effects, geometric distortions, and terrain variations to produce accurate and visually appealing representations of the Earth. Carbon Mapper's operational workflow includes the use of Planet's Planetscope 5 meter resolution visible band images that are updated globally on a monthly cadence. In cases where Tanager's 30 meter spatial resolution is not sufficient to clearly identify the sector/facility type, Carbon Mapper may request high resolution (

#### L2B Atmospheric retrievals

L2B products consist of orthorectified full-strip atmospheric retrieval images derived by the Carbon Mapper data platform from L1B calibrated radiance files to retrieve column or concentration length CH<sub>4</sub> enhancements (units ppm-m) in the strong methane SWIR band between 2200-2400 nm and concentration length CO<sub>2</sub> enhancements

between 1900-2100 nm. We use algorithms that build on experience gained from previous airborne surveys with the AVIRIS-NG and GAO imaging spectrometers and analysis of EMIT data. Specifically, the Carbon Mapper operational workflow uses a column-wise matched filter algorithmthat seeks an estimate for concentration length enhancement of CH<sub>4</sub> or CO<sub>2</sub> for each observed spectrum ((Thompson et al., 2016; Thompson et al., 2015).

#### **Plume Detection**

A point source is defined as the geographic location from which emissions originate that results in a highly concentrated plume of CH<sub>4</sub> or CO<sub>2</sub> gas in the atmosphere. Plumes are an excess mass of concentration in the atmosphere produced by a specific source. Plumes from point sources are a subset of a broader class of CH<sub>4</sub> or CO<sub>2</sub> enhancements that may occur anywhere in the atmosphere as a result of point source and/or diffuse area sources that may or may not be co-located with the enhancements (e.g., a "cloud" of enhanced CH<sub>4</sub> can appear in the atmosphere some distance downwind of the actual source). This is a critical concept: *not all observed atmospheric enhancements are the result of a point source emission nor can those enhancements be reliably attributed to a specific emission source*. Therefore, Carbon Mapper point source detection and quality control procedures require that any detected atmospheric plume must be related to a credible point source on the earth's surface before reporting. Any observed enhancements that fail to meet quality control (QC) checks are noted for potential follow-up study but do not result in published plumes or emission rate estimates.

The Carbon Mapper point source detection process relies on concentration retrievals (CH<sub>4</sub> and CO<sub>2</sub> band images), visible red-green-blue (RGB) imagery from various observing systems, GIS data sets, and meteorological data. The process begins with automated application of CH<sub>4</sub> and CO<sub>2</sub> retrieval algorithms to every calibrated radiance strip image generated by a satellite or airborne sensor. This results in grayscale CH<sub>4</sub> and CO<sub>2</sub> band images that first undergo strip image level QC review by human analysts. This review includes determination of systematic issues affecting the entire strip image, including retrieval processing problems, atmospheric artifacts (high haze, clouds, smoke, etc), geolocation issues, or excessive noise. Each image's CH<sub>4</sub> and CO<sub>2</sub> band images are then reviewed to detect potential point source plumes along with geolocation of their likely origins.

#### **Plume Segmentation**

575

Following detection, Carbon Mapper implements an automated plume segmentation and delineation process on identified and geolocated plumes. This process separates the background from enhanced CH<sub>4</sub>/CO<sub>2</sub> pixels to create a masked plume boundary that is used for mass and emission quantification.

#### **Plume Emissions Quantification**

For emissions quantification, we apply the Integrated Mass Enhancement (IME) approach, which calculates the excess mass in units of kilograms emitted to the atmosphere from a source (Thompson *et al.* 2016). We calculate an emission rate Q using the IME, plume length, and surface wind speed.

$$IME = \alpha \sum_{i=1}^{P} \Omega_i A_i \quad (2)$$

Where *i* refers to a single plume pixel, *P* is the number of pixels in the segmented plume mask,  $\Omega$  is the concentration

Where *i* refers to a single plume pixel, *P* is the number of pixels in the segmented plume mask,  $\Omega$  is the concentration enhancement of that pixel,  $\alpha$  is a unit conversion scalar (from ppm-m to kg/m<sup>2</sup>), and A is the area of that pixel (m<sup>2</sup>). We calculate an emission rate Q using the following relationship (Duren *et al.* 2019):

$$Q = \frac{IME}{L}U \quad (3)$$

Where U is the 10-m wind speed (m/s) and L is the plume length (m). Here U is taken from the HRRR 3km, 60 minute reanalysis product for observations within the U.S. and the ECMWFIFS 9 km product outside the U.S. Forecast versions of these products may be used for initial quick-look processing given standard latencies in receiving reanalysis products. In Equation 3, L is estimated as the maximum distance along the segmented plume's convex hull. For plumes covering large spatial distances, we impose a distance constraint such that the segmented plume mask is clipped to not exceed a 2500 m radial extent from the origin of the plume. Therefore,  $L = \min{\max(\text{hulldist}), 2500\text{m}}$ . The IME (Equation 2) is also only calculated within this clipped plume mask. This clipping procedure is employed to

reduce bias that may affect IME quantification due to differing surface and meteorological conditions across large plumes, intermittency of the emission rate of the source, and to limit potential merging of multiple plumes downwind of their sources. Uncertainty quantification is described in Supplemental Information section SI-4.

#### **Quick look products**

Carbon Mapper's workflow generates quick-look data products with a mean latency of 

#### 3 Tanager-1 commissioning: demonstration of key capabilities

Here we present an initial demonstration of key system capabilities using observations acquired during the first seven months of Tanager-1 operations following instrument activation. After a month of spacecraft and instrument initialization and checkout, a First Light Campaign was conducted between September and December 2024. During this time the satellite orbit was gradually lowered from an average altitude of about 510 km to 430 km – resulting in an instrument FOV and spatial resolution that was initially 6-26% larger than that planned for the final operational orbit. The temporally coarser spatial resolution translated to CH<sub>4</sub> and CO<sub>2</sub> detection limits somewhat higher than expected for nominal operations. During the First Light Campaign, imaging was limited to about 1 observation per orbit on average, most observations were conducted in the Standard (lowest) sensitivity imaging mode, and the length of each image was significantly shorter than available in nominal operations. Despite these limitations, this provided an opportunity to exercise the critical satellite subsystems and enabled a preliminary assessment of the end-to-end performance of the Carbon Mapper emissions monitoring system. Starting in January 2025, Tanager-1 began a multi-month transition to steady state operations, with a steady ramp-up in image size and number of daily observations as well as increased use of Maximum sensitivity imaging mode. This paper includes some early results from this transition phase.

Figure 4 shows a first-light hyperspectral data cube from a Tanager-1 observation of Karachi, Pakistan acquired on September 16, 2024, one month after launch. The first CH<sub>4</sub> plume detected by Tanager-1 in that image was attributed to a known waste dump. The estimated emission rate is  $1224 \pm 221$  kgCH<sub>4</sub>/h. The plume was detected and quantified within 12 hours of the observation. Subsequent analysis of over 1400 plumes detected between Feb 1 and April 1, 2025 indicate good performance against our 72 hour data latency requirement: median 10 hours, mean 34 hours between Tanager image acquisition and Carbon Mapper plume detection. As an illustration of Tanager's broader hyperspectral imaging utility including CH<sub>4</sub>, CO<sub>2</sub> and multiple other environmental variables, Fig. 5 plots the TOA reflectance across the full VSWIR spectral range for 3 pixels in the Karachi image.

**Figure 4.** (Left) Hyperspectral data cube from Tanager-1 first light observation of Karachi, Pakistan on September 16, 2024, one month after launch. (Right) The first  $CH_4$  plume detected by Tanager-1 at a waste dump within the same image, overlaid on a non-contemporaneous Planet high resolution Skysat visible image. The estimated emission rate is  $1600 \pm 300 \text{ kgCH}_4/\text{h}$ . The plume was detected and quantified within 12 hours of the observation.

Figure 5. Plots of full VSWIR TOA reflectance from 3 pixels containing buildings [1], vegetation [2], and water [3] in the image in Figure 11 illustrating Tanager's hyperspectral sensitivity to CH<sub>4</sub>, CO<sub>2</sub>, and multiple other environmental variables.

Between September 16, 2024 and August 15, 2025, Tanager-1 imaged nearly17,000 scenes distributed globally (Fig. 6, top panel), about 35% of which were from the priority CH<sub>4</sub> tasking deck. The remaining scenes were of diverse land or ocean sites or island chains in support of non-trace gas hyperspectral applications or routine radiometric calibration where one would not expect to see strong CH<sub>4</sub> or CO<sub>2</sub> emissions. Additionally, the average daily coverage of Tanager-1 was initially limited to about 22,000 km<sup>2</sup> but since the end of commissioning on January 2025 has dramatically increased to over 200,000 km<sup>2</sup>. As of August 15, 2025, about 5600 CH<sub>4</sub> and about 1200 CO<sub>2</sub> point source emission plumes have been detected in Tanager-1 scenes. The geographical distribution of detected plumes is shown in Fig. SI-5. Roughly 85% of those plumes were detected after Commissioning was completed on January 31, 2025. Many of the scenes to date were collected in at initial altitudes as high as 510 km, resulting in coarser spatial resolution and higher plume detection limits. In most cases, Standard (1x8) Sensitivity imaging mode was used with shorter than normal line lengths (e.g., 

**Figure 6.** Map of the Priority CH<sub>4</sub> tasking deck from September 16, 2024 through August 15, 2025 representing about 35% of 16,952 Tanager-1 scenes collected during that interval. This resulted in 6,813 CH<sub>4</sub> and CO<sub>2</sub> plume detections globally. The zoom panels provide two regional examples of the frequency of plume detections within scenes collected to date for the Priority CH<sub>4</sub> deck.

One motivation for Tanager high spatial resolution mapping is to help address the lack of granular CH4 and CO2 data in the tropics and other persistently cloudy regions that otherwise can remain unobserved for months or years. Frankenberg et al. (2024) analyzed global observations by the Sentinel-2 satellite with 10 meter spatial resolution and 5 minute mean revisit interval to show that satellites with a 30 meter spatial resolution and wide area mapping should achieve median cloud-free (< 0.1% of a pixel) coverage in the Amazon of 10% and 20% during the rainy and dry season, respectively. In contrast, satellites with 1 km spatial resolution would achieve cloud-free yields of only 0.5% and 2% for the wet and dry seasons and satellites with > 2 km spatial resolution would achieve < 0.1% cloud free yields. For reference, the current premier global CH<sub>4</sub> flux mapping satellite (Sentinel-5p/TROPOMI) has a nadir spatial resolution of about 7 km. To evaluate Tanager performance in tackling cloudy images, we targeted several oil and gas basins, landfills and coal mines in the tropics. Figure 7 shows an example of a cloudy image acquired by Tanager-1 in February 2025 in Venezuela (panel A). Carbon Mapper detected a CH<sub>4</sub> plume from an oil and gas facility with roughly 120 meters separation from the nearest cloud (panels B and D). A PlanetScope 5 meter resolution visible image of the same area the same month with no clouds clearly shows the oil and gas facility (panel D). The emission rate estimate from this cloudy scene is consistent with emissions observed by Tanager-1 at the same location under cloud free conditions on three other dates. Tanager-1 has successfully detected similar plumes at multiple sites across the tropics, demonstrating the promise of sustained high-resolution mapping of these critical regions.

Another major design driver for the Tanager instrument is to be robust to challenging observational scenarios including high latitude regions such Russia and Canada where low sun elevation angles and low SWIR albedos from snow covered surfaces can impact SNR and degrade CH<sub>4</sub> and CO<sub>2</sub> detection limits. To evaluate Tanager-1 performance in these conditions we conducted observations of representative high latitude oil and gas production regions during the northern hemisphere winter. Figure 8 shows the results of two such images in Russia in February at 55 degN and 66 deg N latitude. In each case, the ground is covered in snow and solar zenith angles exceeded 70 degrees, however Tanager-1 detected multiple CH<sub>4</sub> plumes.

Figure 7. Tanager-1's 30 meter spatial resolution enables detection of CH<sub>4</sub> plumes in perpetually cloudy regions such as the tropics. In this example of a cloudy image of Venezuela (panel A), Carbon Mapper detected a CH<sub>4</sub> plume from an oil and gas facility within a roughly 700 meter gap between clouds (panel B). The plume mask used to calculate an IME and emission rate does not overlap the cloud (panel C). A PlanetScope 5 meter resolution visible image of the same area in February 2025 with no clouds clearly shows the oil and gas facility (panel D). Three other cloud free observations by Tanager-1on different dates show declining but consistent emissions from this site, averaging  $5600 \pm 500 \text{ kgCH}_4/\text{h}$  (panel E). The basemap overlaid by the Tanager image in panel A is ©Mapbox, ©OpenStreetMap, and ©Maxar.

Figure 8. Two examples of Tanager's CH<sub>4</sub> detection capabilities for challenging high latitude winter images in Russia due to large solar zenith angles and low SWIR albedo due to snow covered surfaces. (left) 20 x 20 km<sup>2</sup> subset of a Tanager-1 image at 55 deg N latitude with 13 CH<sub>4</sub> plumes detected on Feb 23, 2025 at 05:51:06 UTC. Emission estimates for the plumes in this image range from about 400 to 2500 kgCH<sub>4</sub>/h. (right) 24 x 24 km<sup>2</sup> subset of a Tanager-1 image at 66 deg N latitude with 6 CH<sub>4</sub> plumes detected on Feb 26, 2025 at 06:54:41 UTC. Emission estimates for plumes in this image range from about 670 to 5000 kgCH<sub>4</sub>/h. In both figures the small letters denote individual plumes. The basemap overlaid by the Tanager image in each case is ©Mapbox, ©OpenStreetMap, and ©Maxar.

As described in section 2.3, Tanager is also designed to detect CH<sub>4</sub> emissions over ocean surfaces which are dark at SWIR wavelengths. To evaluate Tanager glint-mode performance we conducted several multiple sun-glint observations of selected offshore oil and gas production basins. Figure 9 shows a Tanager-1 detection of a CH<sub>4</sub> plume from an oil and gas platform in the Moho Nord block off the coast of the Republic of the Congo. We intend to scale up glint-mode observations of other offshore production areas in the second year of Tanager-1 operations.

Another key objective of Carbon Mapper's observing strategy is to routinely monitor high emission point sources to assess their variability and persistence. We have tested Tanager-1's ability to track individual super emitters with a regular sample cadence. Figure 10 shows one such example: a time-series of CH<sub>4</sub> plumes detected by Tanger-1 at a persistently emitting oil and gas production site in Algeria. Tanager-1 observations occurred on a roughly monthly basis on average from October 2024 through June 2025. The persistent but variable emissions exhibited by this source are not unusual. Globally, roughly 30% of the CH<sub>4</sub> sources detected multiple times by Tanager-1 to date are at least 50% persistent. This includes all sectors – oil and gas, coal, waste and agriculture – some of which are more or less prone to intermittent emissions.

**Figure 9.** Ocean "glint mode" detection of a CH<sub>4</sub> plume from an oil and gas platform in the Moho Nord block off the coast of the Republic of the Congo. CH<sub>4</sub> image derived from a Tanager-1 observation on March 10, 2025 at 10:19:13 UTC. The estimated emission rate is  $2322 \pm 254$  kgCH<sub>4</sub>/h. The basemap in the inset zoom view is from a Planet SkySat visible image (70 cm resolution) acquired March 16, 2025, providing clear attribution of the likely source origin. The basemap overlaid by the Tanager image in the right figure is ©Mapbox. ©OpenStreetMap, and ©Maxar.

To evaluate Tanager's ability to detect CH<sub>4</sub> and CO<sub>2</sub> point sources simultaneously, we observed some of the world's larger cities and industrial regions that host fossil energy production, electricity generation and refineries. This resulted in numerous individual images where multiple CH<sub>4</sub> and CO<sub>2</sub> plumes were detected. Figure 11 is one such example. Tanager-1 imaged Bahrain on April 1, 2025, revealing 8 CH<sub>4</sub> plumes from oil and gas operations and 2 CO<sub>2</sub> plumes from gas fired power plants. The distribution of plume sizes and shapes reveals both the diversity of emission rates and surface wind fields that are common to many regions.

**Figure 10.** Screenshot from Carbon Mapper's public data portal showing a time series of CH<sub>4</sub> plumes detected by Tanger-1 at a persistently emitting oil and gas production site in Algeria. Tanager-1 observations occurred on a roughly monthly cadence from October 2024 through June 2025. Basemap image ©Mapbox, ©OpenStreetMap, and ©Maxar.

Tanager-1 commissioning also provided an opportunity to conduct some leak detection and repair pilot efforts. Figure 12 shows a methane plume from a leaking oil and gas gathering pipeline that was detected in the Texas Permian basin from a Tanager-1 observation on Oct. 9, 2024 with an estimated instantaneous emission rate of 7100  $\pm$  1100 kgCH<sub>4</sub>/h. After Carbon Mapper notified federal and state agencies the next day, the leak was reported to be voluntarily repaired by the operator. A subsequent Tanager observation on Oct. 24 detected no methane at that location. Analysis of contemporaneous AVIRIS-3 aerial surveys of the Permian on October 1, 9 and 10 reveal high emissions at the same location in all 7 observations (in addition to the Tanager detection), indicating a persistent source with an average emission rate of 4200  $\pm$  500 kgCH<sub>4</sub>/h over at least that 10 day interval. This early demonstration bodes well for Carbon Mapper plans to scale-up data sharing with facility operators and expanded mitigation progress globally.

**Figure 11.** Simultaneous detection of 2 CO<sub>2</sub> plumes and 8 CH<sub>4</sub> plumes in a single Tanager-1 image of Bahrain acquired on April, 2025. The CO<sub>2</sub> plumes are attributed to gas-fired power plants. The CH<sub>4</sub> plumes are attributed to oil and gas production. Basemap image ©Mapbox, ©OpenStreetMap, and ©Maxar.

**Figure 12.** (Left) a large persistent methane plume from a leaking oil and gas gathering pipeline was detected in the Texas Permian basin from a Tanager-1 observation on Oct. 9, 2024 with an estimated instantaneous emission rate of 7100 ± 1100 kgCH<sub>4</sub>/h. Carbon Mapper notified federal and state agencies the next day who informed the operator who reported that the leak was promptly repaired. (Right) a subsequent Tanager observation on Oct. 24 detected no methane. The inset image on the right from Planetscope visible band observations indicated a sudden darkening of the surface within 30 meters of the origin of the methane plume shortly before the first Tanager-1 observation, consistent with a potential condensate release from the pipeline. Basemap image ©Mapbox, ©OpenStreetMap, and © Maxar.

#### 4 Performance predictions and validation







In addition to the various functional demonstrations during Tanager-1 commissioning, a series of experiments were conducted to provide quantitative validation of key performance parameters. Maximum CH<sub>4</sub> and CO<sub>2</sub> performance should occur when Tanager-1 reaches its final target altitude (406 km) and LTDN (1200 hours). Additionally, instrument and detection performance is best assessed by completing a statistically robust number of blinded controlled release experiments sufficient to determine probabilistic detection limits. However, these initial experiments of on-orbit measurements provide strong empirical grounding in our predicted ultimate performance.

## 4.1 Single measurement precision and detection limits

The Minimum Detection Limit (MDL) for CH<sub>4</sub> point sources as a function of measurement precision can be estimated using the method described by Jacob *et al* 2016 as follows:

$$MDL = 2\frac{M_{CH4}}{M_a} \frac{UWp\sigma}{g} \quad (4)$$

Where  $M_a = 0.029$  kg/mol and  $M_{CH4} = 0.016$  kg/mol are the molecular weights of dry air and methane, p is the dry atmospheric surface pressure (typically about 1000 hPa), and g = 9.8 m/s<sup>2</sup> is the acceleration of gravity, U is the wind speed in m/s, W is the pixel size in meters.  $\sigma$  is the single measurement precision or the ability to detect a localized enhancement of CH<sub>4</sub> relative to the average local background (assumed here to be 650 mmol/m<sup>2</sup>). A similar approach can be used to estimate the MDL for CO<sub>2</sub> point sources. This approach assumes a plume detection at the level of one or two contiguous pixels. This represents a theoretical minimum case which is generally insufficient for robust plume detection in practice, where our QC procedures generally require evidence of multiple pixels. However, it can serve as a useful benchmark and simple method for relating instrument and retrieval performance to detection. We demonstrate this empirically in section 4.2. Ultimately the more valuable metric to

assess true detection is derived probabilistically through comparison of satellite detection to a variety known releases rates (e.g., Conrad et al., 2023).

Instrument spectral performance (sampling and FWHM) and radiometric performance (SNR) are the primary constraints on σ. For Tanager, spectral sampling (5nm) and FWHM (5.5nm in the SWIR bands) were set by instrument design and alignment as described in section 2. SNR is a more complex function of the instrument design (optical throughput, read noise, etc) and operation (effective integration time) as well as environmental factors such as solar zenith angle, surface albedo and various atmospheric variables. Prior to Tanager-1 launch we used the lab-measured instrument spectral (Fig. SI-2) and radiometric (Fig. SI-3) performance to generate theoretical predictions of the single measurement CH<sub>4</sub> precisions for each imaging mode (Table 4). We simulated a top-of-the atmosphere radiance spectrum for 35 degree solar zenith angle, 25% albedo using the MODTRAN6 radiative transfer model, and then applied the measured Tanager instrument noise. We applied the optimal estimation concentration retrieval algorithm IMAP-DOAS (Frankenberg *et al.* 2005), which provides single-sounding posterior precision for a retrieved column-averaged CH<sub>4</sub> or CO<sub>2</sub> column concentration. We then applied equation 4 and an assumed wind speed of 3 m/s and 30 meter pixel size to calculate the predicted CH<sub>4</sub> MDL for each imaging mode (Table 4).









However, a more direct measure of  $\sigma$  with the as-built system can be obtained empirically by plotting the standard deviation of background CH<sub>4</sub> in units milli-moles/m<sup>2</sup> across over full image strips as a function of scene-averaged surface albedo for Tanager observations spanning a wide range of solar zenith angles. Figure 13 summarizes the geographical distribution of Tanager-1 observations between September 16, 2024 and July 30, 2025 for scenes that were at least 75% cloud free. Of the roughly 4200 scenes shown here, 3950 were in Standard sensitivity mode and 278 were in Maximum sensitivity mode. Some scenes exhibit highly variable albedo due to strong surface heterogeneity (e.g., urban landcovers), however scene-averaging over a large population allows a preliminary estimate of how noise generally relates to environmental conditions in Tanager observations. Figure 14 provides a preliminary empirical assessment derived from the Maximum and Standard sensitivity observations in Fig 13 where  $\sigma$  is calculated as the standard deviation of non-plume background CH<sub>4</sub> within an image strip using the Columnwise Matched Filter algorithm that is the core of Carbon Mapper's operational data workflow. This confirms the predicted 50% reduction in noise for Maximum sensitivity mode. The Tanager instrument design is based on a reference observation with 25% albedo and 45 degree solar zenith angle. To evaluate the precision for that case we can filter the scenes in Fig 13 to include albedos ranging from 0.2 to 0.3 (average 0.25) and solar zenith angle ranging from 40 to 50 degrees (average 45 degrees). The mean  $\sigma$  for the resulting 164 Standard sensitivity scenes is 12.09 mmol/m<sup>2</sup> (1.86%); assuming 650 mmol/m<sup>2</sup> background. For the resulting 6 Maximum sensitivity scenes the mean  $\sigma$  is 6.11 mmol/m<sup>2</sup>, (0.94%). Those values are equivalent to a CH<sub>4</sub> MDL of 64 and 126 kgCH<sub>4</sub>/h, respectively, for a 30 meter pixel size and 3 m/s wind speed (Table 5). This is in good agreement with pre-launch predictions, particularly considering that the former used the higher sensitivity IMAP-DOAS algorithm. Repeating this exercise for CO<sub>2</sub> and assuming 109,030 mmol/m<sup>2</sup> backgrounds we estimate single measurement precisions of 0.29% and 0.50% and MDL of 10,078 and 18,994 kgCO<sub>2</sub>/h, respectively, for Maximum and Standard sensitivity modes. As discussed in section 2, our methane MDL requirements were derived from a goal of a 90% POD of 100 kg/CH<sub>4</sub> which should be achievable with the as-built Tanager precision and spatial resolution however completion of additional empirical field testing will be necessary for confirmation.

**Table 4.** Pre-launch predictions of single measurement precision and MDL for CH<sub>4</sub> point sources by imaging mode using measured instrument performance and modeled radiances for plumes assuming 25% albedo, 35 degree solar zenith angle, 3 m/s wind speed and 406 km orbit altitude. Additionally, a prediction of CH<sub>4</sub> 90% POD is derived from a linear relationship between MDL and POD observed in empirical field testing of similar airborne instruments.

| Imaging Mode<br>(Sensitivity) | CH <sub>4</sub> single measurement precision (%) | CH <sub>4</sub> MDL<br>(kg/h) | CH <sub>4</sub> 90%<br>POD (kg/h) |
|-------------------------------|--------------------------------------------------|-------------------------------|-----------------------------------|
| Maximum                       | 0.94 %                                           | 63                            | 90                                |
| High                          | 1.07 %                                           | 73                            | 100                               |
| Medium                        | 1.35 %                                           | 92                            | 125                               |
| Standard                      | 1.99 %                                           | 135                           | 180                               |

Figure 13. Geographical distribution of roughly 4200 scenes imaged by Tanager-1 between September 2024 and July 2025 that were at least 75% cloud free and used for empirical assessment of single measurement precision.

**Figure 14.** Empirical evaluation of single measurement precision – estimated as the standard deviation of background noise across each scene as a function of albedo at 2140 nm for the population of scenes shown in Figure 13.

**Table 5.** Single measurement CH<sub>4</sub> precision and MDL for Maximum and Standard imaging modes derived from Tanager on-orbit observations. Empirical precision is calculated from the standard deviation of background CH<sub>4</sub> (assuming 650 mmol/m<sup>2</sup> background) and CO<sub>2</sub> (assuming 109,030 mmol/m<sup>2</sup> background) across entire scenes using the operational CMF retrieval algorithm. This calculation was performed on Tanager-1 scenes with albedos ranging from 20 to 30% (mean 25%) and solar zenith angles ranging from 40 to 50 degrees (mean 45 degrees). This empirical assessment is more conservative than the theoretical pre-launch predictions which used the higher precision IMAP-DOAS algorithm on a simulated image with a smaller solar zenith angle.

| Imaging Mode<br>(Sensitivity) | Mean CH <sub>4</sub><br>Measurement<br>Precision | CH4 MDL<br>(kg/h) | Mean CO <sub>2</sub><br>Measurement<br>Precision | CO2 MDL<br>(kg/h) |
|-------------------------------|--------------------------------------------------|-------------------|--------------------------------------------------|-------------------|
| Maximum (4x8)                 | 0.94%                                            | 64                | 0.29%                                            | 10,078            |
| Standard (1x8)                | 1.86%                                            | 126               | 0.50%                                            | 18,994            |

### 4.2 Validation against independent measurements

Empirical studies (e.g., Sherwin *et al.* 2024, Ayasse *et al.*, 2023) underscore the importance of looking beyond simple analytic predictions of MDL to specify a 90% POD that reflects real world performance over a broader range of conditions. The latter requires a statistically robust set of blinded controlled release tests (e.g., typically > 50 samples which for most satellites can require up to a year to complete when limited to a single test site). While single-blind controlled release testing of Tanager-1 is underway now and anticipated to continue through 2025, we have conducted some initial experiments in the meantime that provide confidence in our pre-launch performance predictions.

915

920

910

During Tanager-1 commissioning phase JPL's AVIRIS-3 aircraft instrument (Green *et al.*, 2022) conducted coordinated under-flights over known high CH<sub>4</sub> emitting regions across the western US, typically at 8 km altitude with about 4.5 meter spatial resolution. The objective of these flights was to provide contemporaneous observations of the same CH<sub>4</sub> sources observed by Tanager-1 including super emitters in oil and gas basins across New Mexico and California. Figure 15 shows good agreement for 20 CH<sub>4</sub> plumes detected by contemporaneous (mean temporal separation 

**Figure 15.** Comparison of Carbon Mapper emission estimates for 20 CH<sub>4</sub> plumes observed by both AVIRIS-3 and Tanager-1 during near-simultaneous overpasses. The median separation between observations was 12 minutes (mean 15 minutes, maximum 37 minutes). The slope and R<sup>2</sup> for an ordinary least squares fit are shown. The error bars represent 1 standard deviation uncertainties in the Tanager-1 and AVIRIS-3 emission rate estimates.

Unblinded controlled release experiments of were conducted between September 21 and October 16, 2024, in Evanston, Wyoming (41.275815, -110.930561), and between November 4 and December 31, 2024, in Casa Grande, Arizona (32.821921, -111.785396). Test setup details are described in SI-5.

Figure 16 shows an example of near-simultaneous (

**Figure 16.** CH<sub>4</sub> retrieval outputs for plumes detected during near-simultaneous observations of a controlled release test in Arizona on November 4, 2024 with Tanager-1 at an altitude of about 500 km at 18:16:42 UTC (left panel) and AVIRIS-3 at an altitude of about 8 km at 18:17:10 UTC (right panel). In this case the AVIRIS-3 image is about 8 times higher spatial resolution than the Tanager-1 image. The images indicate consistent plume shape between the two observations. The geolocation of the methane plume origin from the two observations agreed to within 20 meters. The emission rate estimate for the single Tanager-1 image was  $775 \pm 111 \text{ kgCH}_4$ /h. The mean emission rate from three AVIRIS-3 observations within 10 minutes of the Tanager overpass was  $882 \pm 133 \text{ kgCH}_4$ /h. The mean metered emission rate as reported by the controlled release team corresponding to the three AVIRIS-3 observations was  $859 \pm 49 \text{ kgCH}_4$ /h.

**Figure 17.** Comparison of Carbon Mapper estimated emission rates and metered CH<sub>4</sub> emission rates from the Stanford/U. Michigan controlled release team for cooperative (non-blind) testing of Tanager-1 at test sites Arizona and Wyoming. The slope and R<sup>2</sup> for an ordinary least squares fit are shown. The error bars represent 1 standard deviation uncertainties in the Tanager-1 emission estimates and metered emission rates. The observations shown here were collected at initial higher orbital altitudes ranging from 430 to 510 km and used both maximum and standard sensitivity imaging modes. These initial experiments were designed to provide an initial evaluation of precision and bias rather than probing detection limits and do not represent final sensitivity.

As an additional check on the single pixel MDLs presented in Tables 4 and 5, we compare Tanager detections to independent metered rates and AVIRIS-3 quantified rates near the predicted Tanager MDL. Figure 18A shows a multi-pixel plume detected by Tanager-1, acquired in Maximum Sensitivity mode, for the lowest unblinded controlled release test with a reported release rate of  $99 \pm 4$  kg/h on December 21, 2024 at 18:24 UTC. Figure 18B shows another plume detected by Tanager-1 in Standard Sensitivity mode in the Permian Basin on October 4, 2024 at 17:48 UTC that was also detected by AVIRIS-3 and quantified by AVIRIS-3 as  $179 \pm 106$  kg/h. In both cases a clear plume, extending well beyond a single pixel is readily visible, suggesting that our MDL predictions are in line with mass-balance noise estimates derived from Equation 4.

Figure 18 Multi-pixel plumes with independent emission rate estimates in the range of the single pixel MDLs described in Tables 4 and 5. Panel A shows a plume detected by Tanager-1 during an unblinded controlled release reported by surface metering as 99 ± 4 kg/h on December 21, 2024 at 18:24 UTC. Panel B shows a plume detected in the Permian basin that was observed near-simultaneously by AVIRIS-3 and quantified by AVIRIS-3 as 179 ± 106 kg/h on October 4, 2024 at 17:48 UTC.

## **5 Summary**

We have described the design and observational strategies of the Carbon Mapper emissions monitoring system and provided an initial validation of the performance of Planet's Tanager-1 satellite through coordinated field measurements. We also demonstrated a range of key functional capabilities that were exercised during commissioning spanning observation, plume detection, quantification and rapid reporting. These empirical results indicate that Carbon Mapper and on-orbit Tanager performance is meeting our performance requirements, laying the foundation for further operational scale-up as more satellites are launched. Future papers will provide additional details on Tanager calibration and validation procedures, additional quantitative demonstration of CH<sub>4</sub> estimation accuracy and probabilistic detection limits through blinded controlled release experiments, and further exploration of observing system completeness informed by actual operational experience.

The Carbon Mapper emission monitoring system is ultimately designed to detect, quantify and track 90% of the world's high emission CH4 and CO2 point sources. Meeting that target will likely require a constellation of 10 or more Tanager satellites, because of observing system completeness demands for increased spatial coverage and sample frequency. Meanwhile, the planned interim constellation of four Tanager satellites is predicted to deliver about 60% completeness for super-emitter detection globally and much higher completeness (approaching 100%) for selected regions. We estimate that this interim capability could enable the detection of about 56 TgCH4/year in super-emitter emissions if CH4 point sources above 100 kgCH4/h contribute 10%, 30%, 50% and 50%, respectively, to the agriculture, oil and gas, coal production and waste management sectors globally using bottom-up methane inventories for those sectors (Saunois *et al.* 2025). While those high emission point sources likely only constitute about 20% of the global anthropogenic methane budget, they are also good candidates for expedited mitigation given those super emitters would be limited to a few thousand sites globally (compared to the millions of facilities that contribute the remaining methane flux including distributed area sources). Given the Global Methane Pledge of reducing methane emissions by 30% by 2030, the ability to expedite action on mitigating methane super emitters could be an important component of the broader portfolio of mitigation programs this decade.

Beyond offering mitigation guidance for methane super-emitters, the Carbon Mapper emissions monitoring system is designed to improve quantitative understanding awareness of high emission CH<sub>4</sub> and CO<sub>2</sub> point sources at 30 meter resolution for key regions around the globe through improved monitoring of the cloudy tropics, high latitudes and offshore oil and gas infrastructure – areas that have traditionally been challenging to observe. In doing so, our system serves as a key component in the growing multi-scale, tiered observing system for CH<sub>4</sub> and CO<sub>2</sub> emissions including other point source imagers as well as area flux mapping satellites.

#### **Data Availability**

- To aid in expanded global awareness and data accessibility, Carbon Mapper publishes CH<sub>4</sub> and CO<sub>2</sub> data for all plumes detected by our system as quickly as 30 days following each observation, including quality-controlled retrieval outputs, plume images, coordinates, emission rates, uncertainties and attribution to source type. All Carbon Mapper CH<sub>4</sub> and CO<sub>2</sub> data is available for viewing and download via the Carbon Mapper public data portal (<a href="https://data.carbonmapper.org">https://data.carbonmapper.org</a>) and API (<a href="https://api.carbonmapper.org/api/v1/docs">https://data.carbonmapper.org</a>) and API (<a href="https://api.carbonmapper.org/api/v1/docs">https://api.carbonmapper.org/api/v1/docs</a>). Carbon Mapper documents including our Data Product Guide, Algorithm Theoretical Basis Documents, and Quality Control Description Document are available in the Technical Resources section of our website
- (<a href="https://carbonmapper.org/resources/technical-resources">https://carbonmapper.org/resources/technical-resources</a>). Additionally, rapid access (within 72 hours) to Carbon Mapper quick-look methane products derived from Tanager is available from Planet for subscribers. Data from the controlled release tests referenced in this paper is available at the following repository <a href="https://doi.org/10.25740/qh001qt3946">https://doi.org/10.25740/qh001qt3946</a>.

### **1045** Author Contributions

RD conceptualized and acquired funding for this work. RD, JN, MK, JG, JM, PG, and RG supervised key program elements. RD, DC, AA, KH, AD, TS, JK, KO, AT, SZ, LS, MG, RN, GB, DRT, and KR developed methodologies and implemented analysis. DJ, ME, FR, TA, AB, and EAK provided validation through independent aircraft underflights and controlled release experiments. RD and DC wrote the draft manuscript, and all coauthors contributed to review and editing.

## **Competing interests**

The authors declare that they have no conflict of interest.

## 1055 Acknowledgements

Foundation, Grantham Foundation for the Protection of the Environment, Bloomberg Philanthropies, and other contributors. Portions of this work were performed at the Jet Propulsion Laboratory, California Institute of Technology, under a contract with the National Aeronautics and Space Administration (80NM0018D0004). The Carbon Mapper emissions monitoring system benefited from over a decade of research and technology development enabled by sustained support of NASA's Earth Science Division including the Carbon Monitoring System and airborne science programs. We thank the operations and data platform teams at Planet and Carbon Mapper for their efforts in managing Tanager observations and data delivery. Finally, we thank the management and staff of JPL and Planet for their support in establishing the unique public-private partnership that has underpinned this work.

Carbon Mapper acknowledges the generous support of our philanthropic donors particularly the High Tide

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
