# Peer review of "The Carbon Mapper emissions monitoring system"

_EGUsphere, 2025_

## Referee Comment (RC1)

This article by Duren et al. presents the ambition of the Carbon Mapper emission monitoring system, its design and its observational strategy (inc. revisit time, coverage and completeness sensitivities to the Tanager satellite constellation size). It also describes the instrument optical design, the satellite platform specifications and technological features. Their data processing pipeline from radiance calibration approach to emission rate quantification is described. First observations and demonstration cases of key Tanager capabilities are showcased. At last, first performance assessment on the commissioning phase data is presented.

I commend Carbon Mapper for their efforts in transparency and the breadth of information shared in this submission to AMT, as well as in the technical documents (ATBD, etc.) made available on their website [URL 1]. However, I also think that the manuscript in its current state attempts to describe too many different aspects of the Carbon Mapper emission monitoring system. Its current scope could very well be covered by three or four self-standing scientific articles (e.g. 1: completeness and Tanager constellation size, 2: on-orbit performance, 3: first Carbon Mapper Tanager data; 4: method description paper). I think this produces an overwhelming read (1295 lines of main text content and 27 Figures) that is spread apart in too many different directions and lacks a clearly defined focus on new scientific materials. Consequently, beyond lengthy descriptions, I think some analyses and discussion of new scientific materials presented in this paper lack depth and sometimes boil down to superficial illustrations.

I think this submission could fit well within the scope of AMT as a scientific article and that it could make a valuable addition to the scientific literature provided significant adjustments are made (1) to deepen the analyses and discussions of new scientific material; (2) to clarify the content and structure around new scientific material and (3) to, at places, adjust the writing style and tone. I elaborate on these significant comments and concerns in the following pages, and provide other significant and minor comments as well.

**Significant comment and concerns on analysis depth and completeness**

*(1) Questions and concerns on the aspects that explore the Tanager constellation size and observation strategy, given Carbon Mapper completeness goals, target lists and instrument performance*

The manuscript discusses the Tanager constellation size and observation strategy, and how it relates to completeness goals in Section 2.1 (lines 213 – 336) and, after a long intermission, in Section 4.3 right at the end of the paper (lines 1204 – 1262).

First, Section 2.1.1 defines the Observing System Completeness C as the product of the detection ($C_D$), spatial ($C_S$) and temporal ($C_T$) completenesses. Important parameters driving these three components are clearly explained but the current manuscript does not provide us with actual results or numbers given their current design parameters and/or design hypotheses.

For $C_D$, given the best available inventories, facility-scale datasets, etc. what is the fraction of covered emissions and/or number of sources covered by the current 90% POD of 90 kg/hr for methane, and the current POD for carbon dioxide? Some numbers are provided for the US before defining $C_D$, have you tried determining the same numbers at global scale?

For $C_S$, can you precisely define what is meant by the phrase "tasking deck"? I find Figure 1 to be poorly informative beyond its illustration purpose: what sort of data is presented on this map? I am seeing different colors, yet no legend, title and/or colorbar is provided to interpret them. Are the colors emission rates? Priority ranks in the list? Please be precise. In addition, I find this global view to be hard to read and I suggest authors to consider zoom panels in key emitting regions (e.g. Permian, Algeria, Russian pipelines, Turkmenistan, Shanxi or Australian coal mines). Explanations on important key features and patterns to be noticed on this map would be appreciated as well.

Why are other sector tasking targets not shown? Even if these are preliminary tasking decks subject to change, I would suggest including all sectors. This could for example help illustrate the areas that provide efficient spatial coverage of targets across many sectors at the same time.

Total imaging area numbers are provided in Figure 1 caption. I would imagine that when setting a target list, spatial optimizations can be found where a large fraction of targets can be covered with a relatively small imaging area, and accessing the final fractions of remote isolated targets requires a disproportionate amount of extra imaging area. I think it would be interesting to provide readers with insight into this trade-off. Providing a Figure showing for different sectors, and sum of all sectors the $C_S$ completeness against imaging area (at global scale and/or per region), and the trade-off points currently selected by Carbon Mapper for their tasking deck would be very interesting (even if they are not the final selected trade-off points).

Finally, reading through the whole paper, I notice that the standard imaging mode 1x8 has a 90% detection threshold of 180 kg/hr (Table 4), nearly twice above the overall >100 kg/hr imaging requirement, while the maximum imaging mode meets the requirement. I imagine that this brings interesting interactions between expected source intensities and respective imaging mode coverages when designing an observation strategy. Unfortunately, no details

are given on how these trade-offs are handled, besides explaining that the different imaging modes provide "flexibility for trading off detection limit versus area coverage" (lines 522-523).

For $C_T$, it is first discussed in Section 2.1.1 and then in Section 4.3. I will discuss both here. The paragraph in Section 2.1.1 clearly explains why it is important to revisit sources often, as some sector can show intermittent emissions, and provides sector-wise intermittency assumption. From the overall C = 90% completeness goal of super-emitters >= 100 kg/hr, these data are then translated into a mean sample frequency goal of "7 days for statistically complete monitoring of key region" (line 334). What is the timeframe over which the C goal is completed with mean 7 day revisit frequency? Please be precise.

Beyond the result itself, this derivation of requirements to optimize $C_T$ is overall devoid of any precise and traceable methodological explanation. How is this derivation done? What is the value of $C_S$ used for this derivation? What is the exact breakdown of the statistical model that has been run to come up with this mean 7 day frequency goal, taking into account a prioritized set of targets with a given spatial distribution, different imaging mode sensitivities and area coverage, known POD given instrument characteristics and meteo, cloud coverage statistics and intermittency assumptions? Has this derivation been done at global scale, or local scale, what are the other assumptions? Please be precise.

Section 4.3 provides more simulation-based results, discussing how the number of satellites impact revisit frequency (Figures 25 and 26), and illustrating spatial coverage achieved given a number of satellites and a temporal window (Figure 27).

I think that the discussion of Figures 25 and 26 would benefit from extra explanations on the shape taken by results, and how authors obtained these results. I can intuitively understand the impact of latitude on overall reduced times to access at higher latitudes compared to the equator, however the four higher peaks showed around ±15° and ±40° are less intuitive to me. I suspect this has to do with accounting for cloud coverage and a prioritized list of targets, but this does not replace a proper explanation of the underlying mechanisms.

Figure 27 is – at last – presented with more methodological details in the main text, listing the inputs to "Planet's internal code for calculating access opportunities and collection optimizations" (lines 1248-1249). I think introducing how this code works is critical to enable readers to better understand the results currently presented in this manuscript, and even more so introduce it *before* the results are presented, not at the very end of the paper. If a reference is available, please provide it. In any case, please provide a digest summary of how it works.

I am quite concerned about the end of Section 4.3 and the Summary: some results are stated in these paragraphs without evidence in the Figures or text:
- *"The simulation shows that four Tanager satellites can observe 93% of the targets on a biweekly cadence and 100% of the targets on a monthly cadence."* (lines 1258-1260)
- *"Meeting that target will likely require a constellation of 10 or more Tanager satellites, because of observing system completeness demands for increased spatial coverage and sample frequency"* (lines 1276-1278)

- *"Meanwhile, the planned interim constellation of four Tanager satellites is predicted to deliver about 60% completeness for super-emitter detection globally and much higher completeness (approaching 100%) for selected regions"* (lines 1278-1281).

The text in Section 4.3 does not allow to evaluate these claims, nor the Figures: Figure 25 and 26 are only related to revisit time without looping back completeness into the calculation. Figure 27 is purely illustrative, showing all red targets falling into blue boxes, but does not contain any temporal dimension allowing to evaluate the above-listed results. I think these results are the most interesting and critical regarding the sizing of the future Tanager constellation, and they must be supported by clear evidence. Following the spirit of a previous suggestion for C_S, I propose authors to remove Figure 27 which is purely illustrative and provide instead – for example – an overview of completeness simulation results showing the achieved completeness (y-axis) over a certain time range (x-axis) obtained with constellations of different satellite numbers (several lines of different colors). This could be done at global scale and/or for selected regions (different panels). Such a Figure would for example allow readers to assess the claims listed above, the results currently provided do not.

Finally, the authors could consider merging Section 2.1.1 and Section 4.3 to improve the clarity of the paper.

**(2) Questions and concerns on the aspects related first on-orbit performance results**

The manuscript also discusses the Tanager satellite on-orbit performance and validation (Sections 4.1 and 4.2).

The empirical noise analysis is interesting and quite well explained. I have several questions/concerns about this effort:
1. How were chosen the scenes used to conduct the analysis? The authors report that 5200 images have already been observed by Tanager (Figure 13 caption), but only "over 300" (line 1071) images were included in this analysis. Not including every image is fine, but the selection process would need to be detailed. Can you also provide a (supplementary) map showing where the scenes come from? Besides, can you please provide the exact numbers of data points included for both imaging modes?
2. I also wonder to what extent images can be only boiled down to a reflectance value and an SZA: how heterogenous are the scenes included in this analysis? For example, I would not expect noise levels to be identical between an homogenous desert image and a heterogeneous urban area in a desert, with dark vegetation spots and bright warehouse rooftops.
3. I think the discussion of the Maximum Sensitivity mode could benefit from more data points, especially also covering the lower albedo surfaces (< 0.1) where I would expect it could yield the best improvements compared to Standard. I realize this is somewhat of a "first-data" paper, but I suggest authors to include more points in this Maximum Sensitivity mode in the revised manuscript if more have been observed.

The validation efforts are well described and results presented clearly. I have only one question and one suggestion:

- The caption of Figure 24 mentions that "most observations shown here were using the standard sensitivity mode". Can the authors be explicit about the exact numbers of each sensitivity mode observations included in the controlled release experiment? Can you also color the points of Figure 24 by sensitivity modes? I especially wonder whether maximum sensitivity translates into lower emission rate uncertainties. This may not always be the case because wind speed plays a major role in emission rate uncertainties but still, these additional details would help readers reflect on these questions.
- I suggest authors to cut the technical gas release details from the main text, and either refer to existing references in the literature, or move the details to supplements to lighten the read of this section.

Finally, I am quite curious about a performance point that has not been mentioned in this section: could the authors consider giving an overview of the manual plume verification statistics and, if they can, how they may possibly depend on e.g. expected target emission intensity, surface heterogeneity surrounding the target, meteorological conditions (wind speed, cloudiness), imaging sensitivity mode, etc.? I think giving a sense of how hard manual verification/detection can be in specific difficult conditions could be a very valuable addition to the scientific literature.

*(3) Questions and concerns on the demonstration of Tanager capabilities based on first observations*

The manuscript illustrates key capabilities of the Tanager satellite by showcasing "usual" CH4 and CO2 plume detections over land (Figure 11, Figure 18), but also in more extreme conditions with scattered clouds (Figure 14), at high latitudes in winter over snow (Figure 15), in glint geometry over water (Figure 16). It also showcases Tanager observation use cases, by showing a temporal monitoring example (Figure 17) or a successful mitigation story (Figure 19).

Regarding data description, I find that Figure 13 top is not very informative beyond showing the location of all Tanager observations. Could authors at least color observations according to their goal (targeting CO2/CH4 emitting regions; non-trace gas hyperspectral applications)? This would help to better compare top and bottom panels of Figure 13, and possibly help identify where CO2/CH4 emissions were targeted without detecting any plume. Regarding non-detection, I wonder if it is possible to at least report the number of observed targets that were expected to show emission plumes and that did not show any over the first months of commissioning?

Regarding Figure 14, I find this plume observation quite surprising. My understanding is that panel B shows a zoom of panel A, where the background imagery within the red frame is the actual surface imagery at the time of the plume observation. If this is indeed the case, the methane plume shown in panel B appears to be partially located above the cloud, at least for what must be lower methane enhancement values located East of the identified source, and at the plume western extremity, where the separate low methane enhancement cluster in the mask seems to be completely located above the cloud. Did Tanager observe a methane plume being emitted and/or transported above clouds? If so, this would be quite a surprising find

that calls for further explanation. Otherwise, can the authors please explain why the plume mask is overlapping cloudy pixels? How are cloudy pixels managed in the Carbon Mapper L2 processing pipeline? Could the plume seemingly appear above the clouds because of small co-location errors between Tanager RGB and SWIR channels? Could the plume be explained by retrieval artefacts caused by the high cloud density? Please elaborate on this surprising case.

Regarding Figure 15, I am quite uncomfortable with the current framing used to compare the Kayrros Sentinel-5P/TROPOMI plume detection dataset – or any Sentinel-5P/TROPOMI plume detection dataset for that matter – against Tanager detection capabilities. The current framing may be read as "these plumes should have been detected with TROPOMI", whereas given the fundamental characteristics of the TROPOMI instrument driving its (higher than Tanager's) detection threshold, the challenging observational conditions that reduce coverage at high latitudes in winter and the custom filtering choices made by plume detection pipelines, it is expected that TROPOMI would miss such emissions (how many plumes did Kayrros detect *in February* above 50°N, between 2019 and 2025?). Tanager detecting these emissions perfectly highlights its *complementarity* to TROPOMI, which is – I understand – what showcasing these observations is about. So, the authors should reformulate the comparison as to not imply that these emissions should have been detected in TROPOMI.

**(4) Additional comments and concerns on the quality of other figures**

The figure quality is very unequal across the manuscript, with e.g. inconsistencies in how CO2 and/or CH4 plumes are plotted. Some figures have no use beyond illustration purposes, and in particular Figure 4 fails to meet basic scientific standards with missing axis labels.

I list my comments for every relevant figure below:
- Figure 1: A colormap or legend to interpret the meaning of colors (see other comments above) should be provided.
- Figure 4: No labels for x- and y-axis! Please provide these elements including physical units.
- Figure 5: Purely decorative, please remove (the whole Section 2.3 could actually be removed, see next significant concern on structure and content).
- Figure 7: Please consider a merge with Table 3, lots of redundancies between these two.
- Figure 8: Purely illustrative, the angles written on the figure are not defined anywhere. Please remove and just provide a plain-English definition of glint geometry.
- Figure 10: Please provide a colorbar with label and units.
- Figure 11: Please provide a colorbar with a label for the left panel.
- Figure 13: Please see comment above about adding extra information.
- Figure 14: Please provide a colorbar and units and a scalebar for panel A.
- Figure 15: Please provide a colorbar, longitude of the observed locations and an indication of the significance of the letter labels of the plumes and the map-pin in the right panel.
- Figure 16: Please provide a colorbar and scalebars.
- Figure 17: Please provide a colorbar.
- Figure 23: Please provide a colorbar.

- Figure 27: This is purely illustrative, please replace with a figure that supports conclusions (see comment above)

**Significant comment and concerns on content**

As explained above, I think that this manuscript is an overwhelming read. It provides a lot of information on the Carbon Mapper emission monitoring system but lacks focus around the main and significant scientific novelties presented in it. I cannot currently recommend the publication of the manuscript without large cuts (or transfers to supplements). Besides, similar themes are inexplicably scattered in different corners of the manuscript (see above for completeness study in Sect. 2.1 and 4.3), so I strongly suggest that these cuts come with structure changes. I provide suggestions of cuts and structure change below.

Cut suggestions:
- Section 2.2, key design parameters driving Tanager detection performance can be quickly summarized in text, and all the non-essential technological details cut from the main text (and moved to supplements if necessary).
- Section 2.3 is a Planet Labs product summary unrelated to the scientific content of this manuscript. It can (or even *has to*) be cut out, and [URL2] can be provided when succinctly describing the Tanager satellite for example. ***An additional reason to remove this section is that it scores high in the iThenticate.com Similarity Report compiled by AMT.***
- Section 2.5.1 provides (to my knowledge) new information on the L1 calibration of the Tanager satellite instruments. However, these methodological elements are not directly related to the new scientific content of this manuscript. I suggest cutting Section 2.5.1 and adding it to the currently provided ATBD documents on Carbon Mapper website and/or move it to the supplements.
- Section 2.5.2 can be heavily summarized focusing on the matched filter and IME methods, and providing a digest explanation of the other steps, namely cloud removal, plume detection, masking, manual verification and publication. References to the already available ATBDs [URL1] can be provided for readers interested in the methodological details. ***An additional reason to heavily summarize this section is that it scores high in the iThenticate.com Similarity Report compiled by AMT.***

I strongly suggest authors to consider a clearer structure centered around the main new scientific materials presented in this manuscript. As an example (already provided above), Sections 2.1 and 4.3 are very related and could be presented at the same place in the manuscript.

**Significant comment on writing style and tone**

Most of my qualms on this aspect are located in Sections 1.2 (lines 155-179), Section 2 introductory paragraph (lines 182-210), Section 2.1.1 introductory paragraph (lines 213 – 228), Section 2.1.2 (lines 338-356), the beginning of Section 2.2 (lines 360-397) and Section 2.3 (lines 455-494); but I also provide examples from other parts of the manuscript as well.

There, I find that the writing style and tone of the manuscript strays away from academic to-the-fact writing and ventures more towards narrative and/or speech related rhetorical devices, more akin to sale pitches than scientific literature.

First, the manuscript insists a lot on the Carbon Mapper team's and the overall experience that has gone into the development of their emission monitoring system, often providing no reference or details on how referring to this experience helps informing the new scientific materials presented in this manuscript. Examples (may be not exhaustive) are listed below:

- Lines 25-28: "*Operational scale-up of our system is centered around a new constellation of hyperspectral satellites enabled by over a decade of sustained instrument technology and algorithm advances and prototyping with aircraft surveys and recent observations by NASA's EMIT instrument on the International Space Station.*"
- Lines 176-179: "*The design of those satellites, along with Carbon Mapper's strategy for emissions monitoring and data platform were informed by nearly a decade of field campaigns and mitigation pilot projects using prototype aircraft sensors including the efforts in California, Colorado, New Mexico, and Pennsylvania.*"
- Lines 339-340: "*In the context of CH4 and CO2 information, we have leveraged nearly a decade of experience translating GHG data into action for diverse stakeholders to define "actionable" to mean data that is precise, timely, useful and reliable.*"
- Lines 360-363: "*Since 2016 our research team has used aircraft imaging spectrometers such as the Airborne Visible/Infrared Imaging Spectrometer (AVIRIS) series and Global Airborne Observatory (GAO) as well as research satellites to refine analysis methods and characterize CH4 and CO2 point source emissions from a range of sectors, resulting in the publication of over 15,000 CH4 plumes to date ([many citations])*"
- Lines 388-392: "*The resulting Tanager instrument design leverages a rich history of previous imaging spectrometer development and refinement by the Jet Propulsion Laboratory over a period of more than 20 years, including instruments such as the Airborne Visible/Infrared Imaging Spectrometer (AVIRIS) series of airborne instruments, the Moon Minerology Mapper, and most recently, the EMIT instrument on the International Space Station.*"
- Lines 720-721: "*We use algorithms that build on experience gained from previous airborne surveys with the AVIRIS-NG and GAO imaging spectrometers and analysis of EMIT data.*"
- Lines 1010-1011: "*Tanager-1 commissioning also provided an opportunity to conduct some leak detection and repair pilot efforts that build on our previous experience with aircraft prototyping.*"

These repeated references to the team's and the overall experience put into Carbon Mapper development are not directly relevant to the new scientific materials presented in this manuscript. I read them as rhetorical devices to convince or persuade readers of the Carbon Mapper team skill and qualification to design and run such an emission monitoring system. I personally need no persuasion nor convincing of this team's experience. To my mind, this is out-of-place in scientific literature, and I strongly suggest that authors remove all or at least most of such devices.

Most of the manuscript sections listed above explain how the ambition of the Carbon Mapper emission monitoring system translates into more precise requirements, trickling down from the triad "complete, actionable, accessible" (line 158, repeated line 186). What is meant by these words is described by repeating the same sentence structure many times:

"*The system design and operations are in turn driven by our priorities of maximizing completeness, actionability, and accessibility.* **We define** *completeness (also called Observing System Completeness) to be [...].* **We define** *actionable to be timely data delivery [...].* **We define** *accessible to mean that data is available [...].*" (lines 184-192).

The "actionable" item in the list is then detailed later in the text using additional repetitions:

"*[...] define "actionable" to mean data that is precise, timely, useful and reliable.* **Precise means** *sufficiently granular geolocation information [...].* **Timeliness means** *delivering data that is within the interest-horizon of a given stakeholder [...].* **Useful means** *data that is delivered in [...]. Finally,* **reliable GHG information means** *data that is accurate [...]*" (lines 339-356).

Other excerpts show similar repetitive patterns:

- Lines 194-210: "***Carbon Mapper's monitoring strategy and the Tanager satellites are designed*** *to optimize completeness. This metric constrains ultimate mitigation potential [...].* ***The Carbon Mapper emissions monitoring system is designed to*** *deliver actionable information - images of emission plumes [...]. Another major* ***design*** *driver is data accessibility, where transparent availability [...]*".
- Lines 375-380: "Practically, this means that each Tanager imaging spectrometer instrument **must have** a detector sensitive to [...]. **It must also have** a small focal ratio [...]. **It must have** a fine spatial resolution [...]. The instrument **must also have** a sufficiently wide swath [...]".

I think that these accumulated repetitions inflate the text unnecessarily, and cast somewhat of a speech-like impression and pompous light on the description of Carbon Mapper system requirements.

I would recommend the authors to revise the text of the listed sections aiming at (1) providing more neutrally phrased information; and (2) synthesizing information redundancy caused by repetitions.

**Other significant comments, related to the literature review**

**GHGSat**
GHGSat is a Canadian company operating – among others – a constellation of high-resolution (25x25 m$^2$) methane sensitive satellites, providing high-resolution observations of methane plumes that allow to (1) quantify emission rates; and (2) pinpoint where emissions come from. GHGSat has been reporting on their methods in scientific literature (e.g. Jervis et al., 2021; McLean et al. 2024) and their observations have been used in different scientific publications (e.g. Varon et al., 2018, 2019, 2020; Maasakkers et al., 2022 or Schuit et al., 2023).

Their observations are in nature comparable to Carbon Mapper's, however GHGSat is not mentioned in the literature review currently provided in the manuscript. Can authors report on GHGSat in their literature review?

**TROPOMI detection threshold**
Lower emission rate TROPOMI plume detections have been reported in the literature than the > 50 t/hr threshold reported by Lauvaux et al. (2022, 2021 is written in the text, please correct). For example, Schuit et al. (2023) provide a methane plume detection threshold of ~8 t/hr. I suggest authors revise the reported TROPOMI methane plume detection threshold or report a range including both references.

**Thermal infrared observations**
AIRS is not the only instrument that can provide mid-tropospheric columns of CO2 and/or CH4. The thermal infrared sounder IASI has been providing similar products since 2006 (e.g. Crevoisier et al., 2009a, Crevoisiser et al., 2009b).

I suggest authors to either include a complete review of GHG-sensitive thermal infrared products or, considering that thermal infrared observations are quite unrelated to high-resolution SWIR-based observations of anthropogenic GHG emissions, to remove the thermal infrared discussion from the literature review.

**References**

[URL 1] https://carbonmapper.org/resources/technical-resources

[URL 2] https://www.planet.com/pulse/modular-extensible-smallsat-platform/

Jervis, D., McKeever, J., Durak, B. O. A., Sloan, J. J., Gains, D., Varon, D. J., Ramier, A., Strupler, M., and Tarrant, E.: The GHGSat-D imaging spectrometer, Atmos. Meas. Tech., 14, 2127–2140, https://doi.org/10.5194/amt-14-2127-2021, 2021.

MacLean, J.-P. W., Girard, M., Jervis, D., Marshall, D., McKeever, J., Ramier, A., Strupler, M., Tarrant, E., and Young, D.: Offshore methane detection and quantification from space using sun glint measurements with the GHGSat constellation, Atmos. Meas. Tech., 17, 863–874, https://doi.org/10.5194/amt-17-863-2024, 2024.

Varon, D. J., Jacob, D. J., McKeever, J., Jervis, D., Durak, B. O. A., Xia, Y., and Huang, Y.: Quantifying methane point sources from fine-scale satellite observations of atmospheric methane plumes, Atmos. Meas. Tech., 11, 5673–5686, https://doi.org/10.5194/amt-11-5673-2018, 2018.

Varon, D. J., McKeever, J., Jervis, D., Maasakkers, J. D., Pandey, S., Houweling, S., et al. (2019). Satellite discovery of anomalously large methane point sources from oil/gas production. *Geophysical Research Letters*, 46, 13507–13516. https://doi.org/10.1029/2019GL083798

Varon, D.J., Jacob, D.J., Jervis, D. and McKeever, J., 2020. Quantifying time-averaged methane emissions from individual coal mine vents with GHGSat-D satellite observations. *Environmental science & technology*, *54*(16), pp.10246-10253.

Maasakkers, J.D., Varon, D.J., Elfarsdóttir, A., McKeever, J., Jervis, D., Mahapatra, G., Pandey, S., Lorente, A., Borsdorff, T., Foorthuis, L.R. and Schuit, B.J., 2022. Using satellites to uncover large methane emissions from landfills. *Science Advances*, *8*(31), p.eabn9683.

Schuit, B. J., Maasakkers, J. D., Bijl, P., Mahapatra, G., van den Berg, A.-W., Pandey, S., Lorente, A., Borsdorff, T., Houweling, S., Varon, D. J., McKeever, J., Jervis, D., Girard, M., Irakulis-Loitxate, I., Gorroño, J., Guanter, L., Cusworth, D. H., and Aben, I.: Automated detection and monitoring of methane super-emitters using satellite data, Atmos. Chem. Phys., 23, 9071–9098, https://doi.org/10.5194/acp-23-9071-2023, 2023.

Crevoisier, C., Nobileau, D., Fiore, A. M., Armante, R., Chédin, A., and Scott, N. A.: Tropospheric methane in the tropics – first year from IASI hyperspectral infrared observations, Atmos. Chem. Phys., 9, 6337–6350, https://doi.org/10.5194/acp-9-6337-2009, 2009.

Crevoisier, C., Chédin, A., Matsueda, H., Machida, T., Armante, R., and Scott, N. A.: First year of upper tropospheric integrated content of CO2 from IASI hyperspectral infrared observations, Atmos. Chem. Phys., 9, 4797–4810, https://doi.org/10.5194/acp-9-4797-2009, 2009.

---

## Author Comment (AC1)

6 September  2025

Subject: response to referee report

We thank both referees for their helpful comments. We have implemented most of the recommended edits.  To address concerns about the manuscript's length and readability, we have reduced the scope of the paper to a first assessment of Tanager-1 performance and moved some detailed supporting text and figures to a Supplemental Information appendix.  The sections on observing system completeness and predicted spatio-temporal coverage of the Tanager constellation were removed and will be covered in greater depth in a subsequent manuscript.

Additional specific responses are summarized below.

Best regards,

Riley Duren on behalf of the co-authors

**Referee 1**

(1) **Observation strategy, given Carbon Mapper completeness goals, target lists and instrument performance.**

*In response to the referee recommendations, we have reduced section 2.1.1 to an introduction of the observing system completeness metric and the $C_D$ parameter. We feel this is sufficient to provide insight into our measurement strategy and design drivers without offering a more in-depth treatment that merits a dedicated manuscript. Similarly, we have removed section 4.3 entirely.*

(2) **Questions and concerns on the aspects related first on-orbit performance results**

I have several questions/concerns about this effort:
1. How were chosen the scenes used to conduct the analysis? The authors report that 5200 images have already been observed by Tanager (Figure 13 caption), but only "over 300" (line 1071) images were included in this analysis. Not including every image is fine, but the selection process would need to be detailed. Can you also provide a (supplementary) map showing where the scenes come from? Besides, can you please provide the exact numbers of data points included for both imaging modes?

*We have clarified the criteria used to select points in figures 20 and 21(now figure 14) and added a map (figure 13) showing the geographic distribution of the scenes used in this analysis. We also clarified the number of data points per imaging mode as requested.*

2. I also wonder to what extent images can be only boiled down to a reflectance value and an SZA: how heterogenous are the scenes included in this analysis? For example, I would not expect noise levels to be identical between an homogenous desert image and a heterogeneous urban area in a desert, with dark vegetation spots and bright warehouse roof tops.

*We have increased the sample size significantly to improve the statistical robustness of this analysis. We also added the following language: Some scenes exhibit highly variable albedo due to strong surface heterogeneity (e.g., urban landcovers), however scene-averaging over a large population allows a preliminary estimate of how noise generally relates to environmental conditions in Tanager observations.*

3. I think the discussion of the Maximum Sensitivity mode could benefit from more data points, especially also covering the lower albedo surfaces (< 0.1) where I would expect it could yield the best improvements compared to Standard. I realize this is somewhat of a "first-data" paper, but I suggest authors to include more points in this Maximum Sensitivity mode in the revised manuscript if more have been observed.

*We have maximized the number of points available in this first data paper from 300 to over 4200 scenes, 278 of which are in Maximum Sensitivity mode. As shown in the new figure 14, this significantly expanded the number of low albedo points (< 0.1).*

• The caption of Figure 24 mentions that "most observations shown here were using the standard sensitivity mode". Can the authors be explicit about the exact numbers of each sensitivity mode observations included in the controlled release experiment? Can you also color the points of Figure 24 by sensitivity modes? I especially wonder whether maximum sensitivity translates into lower emission rate uncertainties. This may not always be the case because wind speed plays a major role in emission rate uncertainties but still, these additional details would help readers reflect on these questions.

*We have revised figure 24 (now figure 17) as suggested and clarified the number of observations in each sensitivity mode. We added a note confirming that maximum sensitivity mode translates to lower emission rate uncertainties due to the improved measurement precision, which indeed is evident in the revised figure.*

• I suggest authors to cut the technical gas release details from the main text, and either refer to existing references in the literature, or move the details to supplements to lighten the read of this section.

*As suggested, we have moved details on the controlled release test to the SI section.*

- Finally, I am quite curious about a performance point that has not been mentioned in this section: could the authors consider giving an overview of the manual plume verification statistics and, if they can, how they may possibly depend on e.g. expected target emission intensity, surface heterogeneity surrounding the target, meteorological conditions (wind speed, cloudiness), imaging sensitivity mode, etc.? I think giving a sense of how hard manual verification/detection can be in specific difficult conditions could be a very valuable addition to the scientific literature.

*While we agree that this would make a valuable addition to the scientific literature, we feel that a proper treatment of those topics is beyond the scope of this paper and would be in conflict with other requests to shorten the manuscript. We are also aware of other plans (e.g., by NIST et al with input from many methane satellite teams) in progress to publish consensus standards on this topic.*

*(3) Questions and concerns on the demonstration of Tanager capabilities based on first observations*

Regarding data description, I find that Figure 13 top is not very informative beyond showing the location of all Tanager observations. Could authors at least color observations according to their goal (targeting $CO_2$/$CH_4$ emitting regions; non-trace gas hyperspectral applications)? This would help to better compare top and bottom panels of Figure 13, and possibly help identify where $CO_2$/$CH_4$ emissions were targeted without detecting any plume. Regarding non-detection, I wonder if it is possible to at least report the number of observed targets that were expected to show emission plumes and that did not show any over the first months of commissioning?

*We have revised figure 13 (now figure 6) to differentiate between our priority $CH_4$ tasking deck and all Tanager scenes acquired through August 15, 2025 (including those acquired for other hyperspectral applications). This provides readers with an indication of where we have focused on potential high emission regions. Additionally, we have added some zoomed in views for representative regions that illustrate the overlap between Tanager scenes and where emission sources have been detected to date. While this offers some qualitative intuition about the distribution of plume detections vs observed areas, we note that there is considerable uncertainty in the distribution of super-emitters globally and we do not have a prior model of the likelihood of their occurrence within a given grid cell. In future, we do plan to publish some quantitative analysis regarding detection rates and completeness once we have acquired a larger number of observations and in particular more samples of key regions to constrain temporal variability.*

Regarding Figure 14, I find this plume observation quite surprising. My understanding is that panel B shows a zoom of panel A, where the background imagery within the red frame is the actual surface imagery at the time of the plume observation. If this is indeed the case, the methane plume shown in panel B appears to be partially located above the cloud.... Did

Tanager observe a methane plume being emitted and/or transported above clouds? If so, this would be quite a surprising find that calls for further explanation. Otherwise, can the authors please explain why the plume mask is overlapping cloudy pixels? How are cloudy pixels managed in the Carbon Mapper L2 processing pipeline? Could the plume seemingly appear above the clouds because of small co-location errors between Tanager RGB and SWIR channels? Could the plume be explained by retrieval artefacts caused by the high cloud density? Please elaborate on this surprising case.

*In figure 14 (now figure 7), the plume does not actually overlay the cloud. The extent of the plume color map in panel B is an artifact of smoothing to aid visual visualization for non-expert users of our public data portal. To address the concern raised here we have added the actual plume mask (panel C) used to calculate an IME and emission rate. Additionally, three other Tanager observations of the same source on different dates including those with cloud free conditions indicate this is a persistent emitting source and our emission rate estimates are consistent across that time series. We have added a new panel to the figure showing that time series.*

Regarding Figure 15, I am quite uncomfortable with the current framing used to compare the Kayrros Sentinel-5P/TROPOMI plume detection dataset – or any Sentinel-5P/TROPOMI plume detection dataset for that matter – against Tanager detection capabilities. ...So, the authors should reformulate the comparison as to not imply that these emissions should have been detected in TROPOMI.

*We thank the referee for flagging this issue and have removed that reference.*

**Additional comments and concerns on the quality of other figures.**

I list my comments for every relevant figure below:
- Figure 1: A colormap or legend to interpret the meaning of colors (see other comments above) should be provided.

*Figure 1 has been removed as part of trimming material.*

- Figure 4: No labels for x- and y-axis! Please provide these elements including physical units.
*Figure 4 (now Figure SI-3) has been amended to provide labels and units.*

- Figure 5: Purely decorative, please remove (the whole Section 2.3 could actually be removed, see next significant concern on structure and content).

*Figure 5 has been removed. We feel that technical information on the Tanager smallsat bus is important reference information but have moved an abbreviated section to the SI supplement.*

- Figure 7: Please consider a merge with Table 3, lots of redundancies between these two.

*We feel that figure 7 helps convey the unique nodding imaging mode that differentiates Tanager from most other imaging spectrometer (mostly pure pushbroom) operations.*

- Figure 8: Purely illustrative, the angles written on the figure are not defined anywhere. Please remove and just provide a plain-English definition of glint geometry.

*We have removed figure 8 and provided a plain English definition.*

- Figure 10: Please provide a colorbar with label and units.

*Figure 10 has been moved to figure SI-4.*

- Figure 11: Please provide a colorbar with a label for the left panel.

*In Figure 11 (now figure 4), the left panel shows an RGB image slice overlaid on the hyperspectral image cube, a standard representation for VSWIR imaging spectrometers. Those layers are false colors and we feel that adding a color scale will not be instructive.*

- Figure 13: Please see comment above about adding extra information.

*Now figure 6, see above response.*

- Figure 14: Please provide a colorbar and units and a scalebar for panel A.

*Color bar added.*

- Figure 15: Please provide a colorbar, longitude of the observed locations and an indication of the significance of the letter labels of the plumes and the map-pin in the right panel.

*Now Figure 8. Color bar added. Reformatted image to remove extraneous labels including the map-pin. The latitude references are solely intended to indicate the high latitude location of the scenes. It's unclear that adding longitude provides useful context in this case.*

- Figure 16: Please provide a colorbar and scalebars.

*Now figure 9. Color bar added.*

- Figure 17: Please provide a colorbar.

*Now figure 10. Colorbar added.*

- Figure 23: Please provide a colorbar.

*Now figure 16. This is a grayscale image of a CH4 retrieval intended to convey consistent plume morphology.*

- Figure 27: This is purely illustrative, please replace with a figure that supports conclusions (see comment above)

*Deleted as part of trimming.*

**Significant comment and concerns on content**

As explained above, I think that this manuscript is an overwhelming read. It provides a lot of information on the Carbon Mapper emission monitoring system but lacks focus around the main and significant scientific novel(es presented in it. I cannot currently recommend the publication of the manuscript without large cuts or transfers to supplements). Besides, similar themes are inexplicably scattered in different corners of the manuscript (see above for completeness study in Sect. 2.1 and 4.3), so I strongly suggest that these cuts come with structure changes. I provide suggestions of cuts and structure change below.

*We have endeavored to streamline and shorten the manuscript as suggested. As described above we have removed much of section 2.1 and all of section 4.3. Those topics will be addressed in a future manuscript. Additionally, we have transferred much of the detailed material to SI section.*

Cut suggestions:
- Section 2.2, key design parameters driving Tanager detection performance can be quickly summarized in text, and all the non-essential technological details cut from the main text (and moved to supplements if necessary).

*Most of this section has been moved to SI section.*

- Section 2.3 is a Planet Labs product summary unrelated to the scientific content of this manuscript. It can or even has to) be cut out, and [URL2] can be provided when succinctly describing the Tanager satellite for example.

*Shortened to key technical parameters and moved to SI section.*

- Section 2.5.1 provides (to my knowledge) new information on the L1 calibration of the Tanager satellite instruments. However, these methodological elements are not directly related to the new scientific content of this manuscript. I suggest cutting

Section 2.5.1 and adding it to the currently provided ATBD documents on Carbon Mapper website and/or move it to the supplements.

*We have moved most of this material to SI section as suggested.*

- Section 2.5.2 can be heavily summarized focusing on the matched filter and IME methods, and providing a digest explanation of the other steps, namely cloud removal, plume detection, masking, manual verification and publication. References to the already available ATBDs [URL1] can be provided for readers interested in the methodological details.

*We have heavily summarized this material in the main body and moved details to SI section as suggested.*

I strongly suggest authors to consider a clearer structure centered around the main new scientific materials presented in this manuscript. As an example (already provided above), Sections 2.1 and 4.3 are very related and could be presented at the same place in the manuscript.

*We have incorporated this feedback in our responses above.*

**Significant comment on writing style and tone**
I would recommend the authors to revise the text of the listed sections aiming at (1) providing more neutrally phrased information; and (2) synthesizing information redundancy caused by repetitions.

*We have carefully reviewed and endeavored to address the comments regarding writing style and tone throughout the manuscript.*

***Other significant comments, related to the literature review***

***GHGSat***
GHGSat is a Canadian company operating – among others – a constellation of high-resolution (25x25 m2) methane sensitive satellites, providing high-resolution observations of methane plumes that allow to (1) quantify emission rates; and (2) pinpoint where emissions come from. GHGSat has been reporting on their methods in scien(fic literature (e.g. Jervis et al., 2021; McLean et al. 2024) and their observations have been used in different scientific publications (e.g. Varon et al., 2018, 2019, 2020; Maasakkers et al., 2022 or Schuit et al., 2023). Their observations are in nature comparable to Carbon Mapper's, however GHGSat is not mentioned in the literature review currently provided in the manuscript. Can authors report on GHGSat in their literature review?

*We thank the referee for flagging this oversight. We have added a new section that offers a broader review of point source imaging satellites where we elaborate more fully on GHGSat and other relevant sensors to provide better context for where Carbon Mapper and the Tanager satellites contribute to the broader ecosystem of satellites.*

**TROPOMI detection threshold**

Lower emission rate TROPOMI plume detections have been reported in the literature than the > 50 t/hr threshold reported by Lauvaux et al. (2022, 2021 is written in the text, please correct). For example, Schuit et al. (2023) provide a methane plume detection threshold of ~8 t/hr. I suggest authors revise the reported TROPOMI methane plume detection threshold or report a range including both references.

*We thank the referee for flagging this and have added the Schuit et al number and citation.*

**Thermal infrared observations**

AIRS is not the only instrument that can provide mid-tropospheric columns of CO2 and/or CH4. The thermal infrared sounder IASI has been providing similar products since 2006 (e.g. Crevoisier et al., 2009a, Crevoisiser et al., 2009b). I suggest authors to either include a complete review of GHG-sensitive thermal infrared products or, considering that thermal infrared observations are quite unrelated to high resolution SWIR-based observations of anthropogenic GHG emissions, to remove the thermal infrared discussion from the literature review.

*We agree and have eliminated the reference to thermal IR sounders.*

---

## Author Comment (AC2)

6 September 2025

Subject: response to referee report

We thank both referees for their helpful comments. We have implemented most of the recommended edits.  To address concerns about the manuscript's length and readability, we have reduced the scope of the paper to a first assessment of Tanager-1 performance and moved some detailed supporting text and figures to a Supplemental Information appendix.  The sections on observing system completeness and predicted spatio-temporal coverage of the Tanager constellation were removed and will be covered in greater depth in a subsequent manuscript.

Additional specific responses are summarized below.

Best regards,

Riley Duren on behalf of the co-authors

**Referee 2**

Here is a list of the parts of the manuscript where I think improvements are needed:

- Abstract: it is very long, and reads more as an executive summary of an internal mission report than as an abstract of a scientific pubilication. I would propose to shorten it substantially, especially in the parts not directly related to findings of this study.

*We have revised the abstract as suggested.*

- Sec. 1.1: these paragraphs provide a review of past and current missions with sensitivity to methane and CO2. However, the part of this section referring to "point source images" is strongly biased towards the instruments and work by the authors. It is striking (and a bit annoying) not to find a single reference to the GHGSat program, which is very similar to Carbon Mapper in terms of observational requirements and capabilities (with a superior performance for GHGSat currently because of their higher number of operating satellites). It is also surprising not to find references to the retrieval and analysis work that has been done (mostly by groups in Europe and China) with other space-based imaging spectrometers, including EnMAP, PRISMA and the AHSI onboard the GF and ZY1 satellites, which are also very similar to the Tanager instruments. I would strongly request the authors to better reflect the international context in their study.

*We have added a new section 1.2 that summarizes the state of the art in point source imaging satellites that includes GHGSat, EnMAP, PRISMA, AHSI and EMIT to provide better context for the contributions of Carbon Mapper/Tanager to the growing ecosystem.*

- Sec. 2: it is also very lengthy: In my opinion, the first two paragraphs read as a new introduction section, sec 2.1.2 does not add meaningful content, L388-397 are redundant with previous contents, and L399-419, L470-480 and L620-655 provide much more detail on the instrument design than what is actually needed to understand Tanager's potential for GHG retrievals. I believe that the whole section would benefit from shortening and focusing on the mission and instrument parameters directly affect

*We have significantly reduced the length of section 2 by moving some material to SI section and removing other material to be covered in a future manuscript.*

- Sec. 2.5.3: please explain how uncertainties in emission rates are estimated

*We have added a section on uncertainty quantification in the SI section*

- Sec. 4.1, MDL: does this MDL analysis only refer to one pixel standing out from the background XCH4 values, as I seem to interpret from Eq. 7? I don't think that you would claim a plume detection if this is only based on a 1-pixel enhancement, but you would need several connected pixels with an enhancement n-times higher than the noise level. Is this correct? If so, I don't this metric can be used as an absolute measure of detection limit, as I think you are doing within this section.

*Equation 7 (now equation 4) provides a first order method for estimating MDL based on 1-2 pixels as outlined in Jacob et al. However, this is not equivalent to saying that we would report a plume detection based solely on a 1-2 pixel enhancement. We explain at multiple places throughout the manuscript that a more robust method for determining detection limit is empirical testing to establish a Probability Of Detection (POD). As discussed, empirical testing to determine POD can take upwards of a year for most satellites so we're not yet able to report that. Instead, as explained we summarize what we think is a robust initial assessment of single measurement precision as a check of whether the instrument and retrieval algorithms are performing as designed and on track to meet sensitivity targets. We do not rely solely on MDL as an absolute measure of detection limit as explained in the text. However, to address this concern we have added the following text and a new figure: "As an additional check on the single pixel MDLs presented in Tables 4 and 5, we compare Tanager detections to independent metered rates and AVIRIS-3 quantified rates near the predicted Tanager MDL. Figure 18A shows a multi-pixel plume detected by Tanager-1, acquired in Maximum Sensitivity mode, for the lowest unblinded controlled release test with a reported release rate of $99 \pm 4$ kg/h on December 21, 2024 at 18:24 UTC. Figure 18B shows another plume detected by Tanager-1 in Standard Sensitivity mode in the Permian Basin on October 4, 2024 at 17:48 UTC that was also detected by AVIRIS-3 and quantified by AVIRIS-3 as $179 \pm 106$ kg/h. In both cases a clear plume, extending well beyond a single pixel is readily visible, suggesting that our MDL predictions are in line with mass-balance noise estimates derived from Equation"*

Other minor comments:
- L151 "of CH4"
*Corrected.*

- L343 "types. And"
*Corrected.*

- Table 1, I miss the GSD parameter
*Added.*

- Figs.3 and 4: they should have a more similar format. Also, axis labels are missing in Fig.4.
*Moved to SI section and corrected.*

- L656 FPA has been defined ealier in the text

*Corrected.*

- L778 and L785: two consecutive definitions of QC

*Corrected.*

- L895 TOA has been defined ealier in the text

*Corrected.*

- L978 "Condo"?

*We have vacated our Condo in the Congo. Seriously, thanks for the catch.*

- L1010-1016: As the authors know, super-emissions in the Permian basin are typically short-lived. I don't think that the 7 t/h source would have been active 15 days after the initial Tanager detection even if it had not been notified.

*The observational evidence suggests that this source persisted for at least 10 days, perhaps longer. This is consistent with analysis of previous observations of super emitters in the Permian. We added the following text regarding this example: "Analysis of contemporaneous AVIRIS-3 aerial surveys of the Permian on October 1, 9 and 10 reveal high emissions at the same location in all 7 observations (in addition to the Tanager detection), indicating a persistent source with an average emission rate of 4200 $\pm$ 500 kgCH$_4$/h over at least that 10 day interval."*

- Fig. 20, 21: ppm·m units should be used for consistency with the other figures and XCH4 maps

*Our experience is that when it comes to representing single measurement precision and noise relative to background concentrations, the convention is to use mmol m-2 and % rather than ppm-m. We feel this offers a more consistent comparison with precision and noise reported elsewhere in the literature.*

- L1210: "for an isolated of interest"

*This section was removed in the interest of reducing the manuscript length.*

- Fig. 25, 26: they should have the same y-axis label; also, please, discuss the TTA peaks in the text.

*This section was removed in the interest of reducing the manuscript length.*

---

## Author Response (AR2)

**15 October 2025**

**Editor responses to Duren et al**

We thank the editor for their helpful comments. We have implemented the corrections requested by the editor as outlined below. Additionally, we made some minor corrections to numerical values in Figure 2 and Table 3 to reflect the latest Tanager satellite specifications. Finally, we confirmed that appropriate credit has been provided for all figures that include images generated by third parties.

Please let us know if there are any follow up questions.

Best regards,

Riley Duren

Figure 16: even if the figure purpose is to illustrate spatial consistency between both plumes, I insist on adding a color bar. One can only assume here that black-to-white corresponds to low-to-high methane enhancement values, but also that the color scale range is identical between both panels, which may not be the case? I would expect that the higher spatial resolution of AVIRIS-3 allows to observe stronger enhancements than with Tanager 30m resolution, but the current grayscale without color bar does not allow to assess that. Also, one has to assume that the spatial extents of both panels are identical in this figure, is this accurate? The caption can be edited to confirm this point.

Response: we replaced Figure 16 with a colorized version including color bar and scale bar.

Internal references: I recommend authors to carefully proof-read their internal references, I could find two discrepancies:

- line 689: "[...] shows the oil and gas facility (panel C)." -> panel D, instead of C currently
- line 936: "Test setup details are described in SI-4." -> SI-5, instead of SI-4

Response: we corrected those issues and carefully proof-read other internal references.

References: added references from Razavi et al. (2009), Wecht et al. (2012) and Worden et al. (2024) are not cited anywhere in the text.

Response: we removed the extraneous references.